



# Impact of ocean vertical mixing parameterization on Arctic sea ice and upper ocean properties using the NEMO-SI3 model

Sofia Allende[1], Anne Marie Treguier[2], Camille Lique[2], Clément de Boyer Montégut[2],
François Massonnet[1], Thierry  Fichefet[1], and Antoine Barthélemy[1]

[1]Earth and Climate Research, Earth and Life Institute, Université catholique de Louvain, Louvain-la-Neuve, Belgium
[2]Univ Brest, CNRS, IRD, Ifremer, Laboratoire d'océanographie physique et spatiale, IUEM, 29280, Plouzané, France

**Correspondence:** sofia.allende@uclouvain.be

**Abstract.**

We evaluate the vertical turbulent kinetic energy (TKE) mixing scheme of the NEMO-SI3 ocean–sea ice model in sea ice-covered regions of the Arctic Ocean. Specifically, we assess the parameters involved in the TKE mixed layer penetration (MLP) parameterization. This ad-hoc parameterization aims to capture processes like near-inertial oscillations, ocean swells, and waves that impact the ocean surface boundary layer, often not well-represented in the default TKE scheme. We evaluate this parameterization for the first time in three regions of the Arctic Ocean: the Makarov, Eurasian, and Canada Basins.

We demonstrate the strong effect of the scaling parameter that accounts for the presence of sea ice. Our results confirm that the TKE MLP must be scaled down below sea ice to avoid unrealistic deep mixed layers. The other parameters evaluated are the percentage of energy penetrating below the mixed layer and the length scale of its decay with depth. All these parameters affect the mixed layer depth and its seasonal cycle, the surface temperature and salinity, as well as the underlying stratification. Shallow mixed layers are associated with stronger stratification and fresh surface anomalies, and deeper mixed layers correspond to weaker stratification and salty surface anomalies.

Notably, we observe significant impacts on sea ice thickness across the Arctic Ocean in two scenarios: when the scaling parameter due to sea ice is absent and when the TKE mixed layer penetration process vanishes. In the former case, we observe an increase of several meters in the mixed layer depth together with a reduction in sea ice thickness ranging from 30 to 40 centimeters, reflecting the impact of stronger mixing. Conversely, in the latter case, we notice that a smaller mixed layer depth is accompanied by a moderate increase in sea ice thickness, ranging from 10 to 20 centimeters, as expected from a weaker mixing. Additionally, inter-annual variability suggests that experiments incorporating a scaling parameter based on sea ice concentration display an increased mixed layer depth during periods of reduced sea ice, which is consistent with observed trends. These findings underscore the influence, through specific parameterizations, of enhanced ocean mixing on the physical properties of the upper ocean and sea ice.





# 1 Introduction

In the last decades, global climate change has strongly affected the Arctic region, leading to a fast decrease in sea ice ex-
tent (Perovich and Richter-Menge, 2009). This phenomenon, together with the increase of openings in the ice pack, modifies
the exchanges between the atmosphere and ocean and hence the fully-coupled atmosphere–sea ice–ocean system in the Arc-
tic (McPhee, 2008). These changes have, in turn, led to alterations in the physical properties of the upper ocean, with implica-
tions for sea ice dynamics and response (Lenn et al., 2022). The upper layer of the Arctic Ocean, known as the Arctic mixed
layer (ML), plays a pivotal role in regulating interactions between the deeper ocean, sea ice, and the atmosphere. Key factors
influencing the Arctic ML include heat, freshwater and momentum fluxes generated by ocean-atmosphere exchanges, currents,
tides and waves (Rabe et al., 2022; Rudels and Carmack, 2022). Observational data from the past few decades reveal changes
in both winter and summer ML across the high Arctic, accompanied by changes in ocean stratification (Cole and Stadler, 2019;
Peralta-Ferriz and Woodgate, 2015). These rapid transformations of the Arctic environment have far-reaching implications for
climate and socio-economic aspects (Ford et al., 2021). Therefore, achieving an accurate representation of the Arctic Ocean
and sea ice in models is essential to understanding and predicting these consequential changes.

Sea ice-ocean general circulation models used in coupled models assessed by the IPCC (Cassotta et al., 2022) exhibit
significant discrepancies in the Arctic ML depth (Allende et al., 2023), which is directly influenced by vertical mass and
momentum exchanges between the upper ocean and sea ice. These small scale vertical processes are parameterized in general
circulation models. The NEMO-SI3 ocean-sea ice model (Madec et al., 2017; Sea-Ice-Working-Group et al., 2020) includes
a vertical turbulent kinetic energy (TKE) closure scheme initially proposed by Blanke and Delecluse (1993). The scheme
is complemented with an additional source of TKE to simulate the effects of near-inertial oscillations, ocean swells, and
waves, specifically known as TKE mixed layer penetration. The TKE mixed layer penetration has been introduced to overcome
summer biases in the mixed layer, which was too shallow in the Southern Ocean (Calvert and Siddorn, 2013; Rodgers et al.,
2014). It redistributes a percentage of the surface TKE below the mixed layer depth. Moreover, the TKE mixed layer penetration
is attenuated in the presence of sea ice. Previous research, such as that by Calvert and Siddorn (2013), has emphasized the
impact of this parameterization on ocean properties in regions without sea ice. Additionally, Heuzé et al. (2015) has underscored
the effect of this parameterization on deep convection in the Southern Ocean. However, the specific influence of TKE mixed
layer penetration under sea ice has not been documented in the literature, although preliminary research carried out within the
ArcticMix project (https://marine.copernicus.eu/about/research-development-projects/2016-2018/arcticmix) suggests that this
influence is large. This is an important issue, because the NEMO-SI3 ocean–sea ice model is extensively used in polar climate
studies (e.g. Goosse et al., 2023; Dong et al., 2023; Docquier et al., 2017; Vancoppenolle et al., 2008) and featured in the IPCC
Assessment Reports.

This study aims to evaluate the impact of changes in the TKE mixed layer penetration in three distinct ice-covered regions
within the Arctic Ocean: the Makarov, Eurasian, and Canada Basins. By varying the parameters within this scheme, we il-
lustrate how ocean and sea ice physical properties respond to alterations in upper ocean mixing. The paper is organized as
follows. Section 2 gives a description of the NEMO-SI3 configuration, parameters, and outputs involved in the study, as well





as the oceanic and sea ice observation data. Section 3 presents a diagnosis of the seasonal and inter-annual variability of the upper ocean and sea ice properties in several regions of the Arctic Ocean, varying the mixing just below the mixed layer TKE parameters. Finally, Section 4 presents concluding remarks and discusses the implications of our work.

## 2   Method

The turbulent kinetic energy (TKE) scheme implemented in NEMO is based on the turbulent closure model developed by Bougeault and Lacarrere (1989) for the atmospheric boundary layer. It was subsequently adapted for an oceanic context by Gaspar et al. (1990) and integrated into the OPA model by Blanke and Delecluse (1993), which is the ocean model component of the NEMO platform. In essence, this TKE turbulent closure model provides a prognostic for the evolution of turbulent kinetic energy ($\bar{e}$) and a closure assumption for turbulent length scales, both necessary for computing vertical eddy viscosity and diffusivity coefficients. The last version of NEMO includes significant modifications introduced by Madec et al. (2017). These changes include considerations such as turbulent length scale adjustments that impose an extra assumption regarding the vertical gradient of the computed length scale (Madec et al., 1997). Additionally, a surface wave breaking parameterization has been incorporated following Mellor and Blumberg (2004), addressing the impact of surface wave breaking energetics. The model now accounts for Langmuir cells using a simple parameterization proposed by Axell (2002) for a $k - \epsilon$ turbulent closure. Furthermore, the TKE turbulent closure has been updated to include mixing just below the mixed layer. Our study focuses on evaluating the influence of this last modification, which we will refer to as TKE Mixed Layer Penetration (MLP). This parameterization has been introduced to address the underestimation of the ML depth (MLD), especially in situations characterized by windy conditions during summer months, as observed in the Southern Ocean (Rodgers et al., 2014). This parameterization aims to account for observed phenomena that impact the density structure of the ocean's planetary boundary layer. These include near-inertial oscillations, ocean swells, and waves, which are not fully captured by the default TKE scheme. The parameterization is activated in NEMO when the parameter nn_etau is set to 1 (and deactivated when nn_etau=0). The TKE $\bar{e}(t,z)$ is then evaluated using the equation: $\bar{e}(t + \Delta t, z) = \frac{\partial}{\partial t}\bar{e}(t,z)\Delta t + \bar{e}_{inertial}(t,z)$, where $\bar{e}_{inertial}$ represents the TKE MLP contribution as:

$$\bar{e}_{inertial}(t,z) = \begin{cases} \chi \, f_r \, \bar{e}_{surf} \, \exp^{-z/h_\tau}, & \text{if } z > 0. \\ 0, & \text{if } z = 0. \end{cases} \tag{1}$$

Here, $z$ is the depth, $f_r$ is the fraction of the surface TKE $\bar{e}_{surf}$ that penetrates into the ocean, $h_\tau$ is the vertical mixing length scale that controls the exponential shape of the penetration, and the degree of this mixing is regulated by the scaling parameter $\chi$ in response to the presence of sea ice. Within the namelist parameters, $f_r$ is explicitly specified by rn_efr with values ranging from 0 to 0.1. It is commonly set to 0.05, which means that 5% of the surface TKE is redistributed below the MLD. The $h_\tau$ parameter can be set as either a uniform 10 m value (nn_htau=0), as a latitude-dependent value, which varies from 0.5 m at the equator to 30 m north to $40°$ latitude (nn_htau=1), or with a different value in the two hemispheres (nn_htau=4), see Fig. 2 of





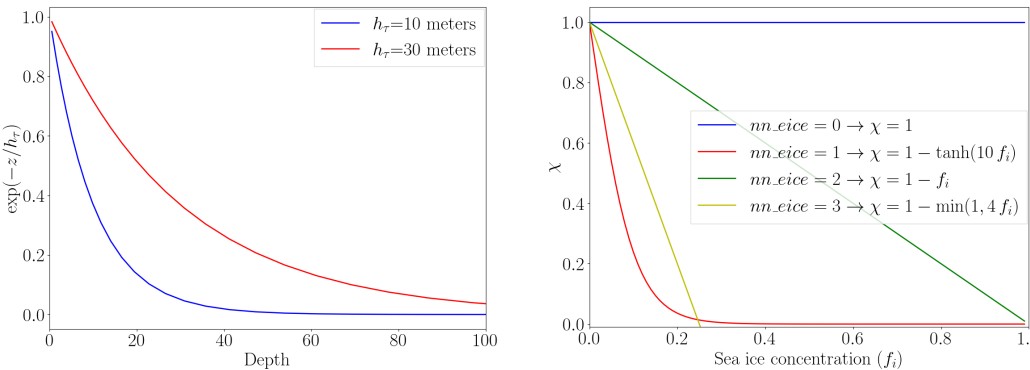

**Figure 1.** Left panel: Exponential penetration as a function of depth, for the two values of the vertical mixing length scale $h_\tau$. Right panel: scaling parameter $\chi$ as a function of the sea ice concentration fraction $f_i$, for the four options of the scaling parameter $\chi$.

Storkey et al. (2018). The penetration scales of 10 and 30 m are illustrated in the left panel of Fig.1; 10 m was found optimal in the north Pacific (Calvert and Siddorn, 2013), while 30 m was required to improve the ML in the Southern Ocean (Storkey et al., 2018). $\chi$ corresponds to the namelist parameter nn_eice. When nn_eice is set to 0, this corresponds to $\chi = 1$, indicating no influence from ice cover; nn_eice set to 1 corresponds to $\chi = 1 - \tanh(10 f_i)$ where $f_i$ represents the sea ice concentration fraction; nn_eice set to 2 corresponds to $\chi = 1 - f_i$; and nn_eice set to 3 corresponds to $\chi = 1 - \min(1, 4 f_i)$ used to suppress TKE input into the ocean when sea ice concentration exceeds 25% (see right panel of Fig. 1).

To carry out this investigation, we utilize the NEMO4.2 version with the SI3 sea ice model, configured with a horizontal resolution of $1°$ (ORCA1). The atmospheric forcing for the ocean is provided by the ERA-5 reanalysis (Hersbach et al., 2020). In our simulations, we configure the setup to exclude salinity restoring under sea ice. The default parameters are configured in NEMO4.2 as follows: rn_efr=0.08, nn_eice=3, and nn_htau=1, constituting the control run for our analysis. Rathore et al. has tuned this specific configuration, differing slightly from the NEMO4.2 reference configuration by increasing the rn_efr parameter from 0.05 to 0.08 to achieve a more realistic MLD in the Southern Ocean. The global ORCA1 configuration has been used in CMIP6 by 6 different groups who made different choices for the TKE MLP parameterization (table 1). CCCma, CMCC and CNRM used the default NEMO settings (note that although the nn_eice parameter had not yet been introduced in NEMO3.4, the $(1 - f_i)$ factor which corresponds to nn_eice=2 was present by default). As noted above, the MOHC group introduced an additional option for the nn_htau parameter (Storkey et al., 2018). EC-Earth and IPSL opted to turn off the MLP parameterization, because doing so improved the Atlantic Meridional Overturning Circulation (AMOC) in their coupled models. These different choices underline the need for further investigation of the MLP parameterization.

We have performed a series of sensitivity experiments to assess the impact of the TKE MLP on sea ice in the Arctic region. Our strategy is to perform long experiments (from January 1960 to December 2022) to assess possible long-term climate impacts. For this reason, the number of simulations is limited, and parameters are modified one at a time. We systematically modify the parameters as outlined below: rn_efr ranges uniformly from 0 to 0.1, with values of 0, 0.005, 0.025, 0.075, and 0.1.





| Group | Model | Version | nn_etau | rn_efr | nn_htau | nn_eice |
|---|---|---|---|---|---|---|
| CCCma | CanESM5.0 | NEMO3.4 | 1 | 0.05 | 1 | 2 |
| CMCC | CMCC-CM2-SR5 | NEMO3.6 | 1 | 0.05 | 1 | 2 |
| CNRM | CNRM-CM6-1 | NEMO3.6 | 1 | 0.05 | 1 | 2 |
| Ec-Earth | Ec-Earth | NEMO3.6 | 0 | - | - | - |
| IPSL | IPSL-CM6A-LR | NEMO3.6 | 0 | - | - | - |
| MOHC | HadGEM3-GC3.1-LL | NEMO3.6 | 1 | 0.05 | 4 | 2 |

**Table 1.** Configuration details of the TKE MLP parameterization in NEMO ocean models from various climate modeling groups participating in CMIP6-OMIP. The table provides information about the modeling group, NEMO version, and the parameters nn_etau, rn_efr, nn_eice, and nn_htau.

Please note that using rn_efr=0 leads to the same results as turning off the MLP parameterization (nn_etau=0), which means $\bar{e}_{inertial}(t,z) = 0$. In addition, we investigate the two values for nn_htau (0 and 1) and the four options for nn_eice (0, 1, 2, and 3). A summary of our experiments is presented in the table. 2.

To investigate the variations in ocean and sea ice properties, we rely on monthly average outputs. Our analysis focuses on the following variables:

- Ocean variables: ocean mixed layer depth (*mldr0_3*), for which the model follow the common threshold density criteria $\Delta\rho = \rho(z) - \rho(z_{ref}) = 0.03\,kg/m^3$, with $z_{ref} = 0.5$ m; seawater potential density vertical profile (*rhop*) in $kg/m^3$, sea water potential temperature vertical profile (*thetao*) in $^{\circ}C$, and the seawater salinity vertical profile (*so*) in pss;

- Sea-ice variables: sea ice concentration (*siconc*), defined as the percentage of the grid cell covered by sea ice, and sea ice thickness (*sivolu*), defined as the total volume of sea ice divided by grid-cell area in $m$.

We evaluate the performance of the NEMO-SI3 model using different sets of observational data and a reanalysis. Specifically, we employ the MLD climatology from IFREMER-LOPS, developed by de Boyer montégut C. (2024). In the following, we refer to this data set as the LOPS climatology. This dataset provides monthly MLD, sea surface temperature and sea surface salinity values across the global ocean at a spatial resolution of 1° by 1°. The climatology is constructed based on approximately 7.3 million temperature and salinity profiles collected at sea between January 1970 and December 2021. This includes data from the ARGO program, the NCEI-NOAA World Ocean Database (Boyer et al., 2018), and Ice-Tethered Profilers (ITP). The MLD is computed on each individual profile using the threshold density criteria $\Delta\rho = \rho(z) - \rho(z_{ref}) = 0.03\,kg/m^3$. This criterion defines the MLD as the depth where the increase of density compared to the density at the reference depth ($= z_{ref}$) exceeds $0.03\,kg/m^3$ (e.g. de Boyer Montégut et al. (2004)). The choice of the reference depth is impactful (Treguier et al., 2023). The LOPS climatology aims at estimating the MLD as the depth of the layer that has been mixed over at least a daily cycle and no more than a few daily cycle. This timescale includes night time convection which is usually assumed to reach at least 10 m depth (e.g. Brainerd and Gregg (1995)). Thus, a 10 m depth is chosen at global scale so as to filter out the possible

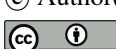



| Parameter | Value | Variable | Value |
|-----------|-------|----------|-------|
| rn_efr | 0 | $f_r$ | 0 |
| rn_efr | 0.005 | $f_r$ | 0.005 |
| rn_efr | 0.025 | $f_r$ | 0.025 |
| rn_efr | 0.075 | $f_r$ | 0.075 |
| rn_efr | 0.1 | $f_r$ | 0.1 |
| nn_eice | 0 | $\chi$ | 1 |
| nn_eice | 1 | $\chi$ | $1 - \tanh(10\,f_i)$ |
| nn_eice | 2 | $\chi$ | $1 - f_i$ |
| nn_htau | 0 | $h_\tau$ | 10m |

**Table 2.** Sensitivity experiments of the TKE MLP parameterization. Each parameter value indicates the change from the control run, along with the corresponding modified variable in Equation 1.

daily stratification in the top few meters (e.g. in the tropics, or in summer mid latitudes). In the Arctic Ocean however, it is expected that we have a quite small diurnal cycle or even no diurnal cycle linked to the solar heat fluxes reaching the ocean, especially when ice is present. The MLD can then have shallower values than the usual 10 m depth minimum (and possibly lasting for several days) as shown in Peralta-Ferriz and Woodgate (2015). For this reason, the MLD has been recomputed here

for the Arctic region, using a more adequate value of $z_{ref} = 5$ meters. Additionally, to study the inter-annual variability of the MLD, we directly computed the MLD from individual ITP data (Toole et al., 2011; Krishfield et al., 2008) using completed missions available at the WHOI website. We compute vertical potential density profiles using the TEOS-10/GSW (Gibbs Sea Water) Python library based on conservative temperature and absolute salinity profiles. We compute the MLD by applying the threshold density criteria, to compare ITP observational data with NEMO sensitivity experiments, where the surface reference

depth for ITP varies from 10 to 0 m, depending on the profile. The ITP data includes data from 2004 until 2019, with the majority of the observations between the years 2007 to 2015. Our study also incorporates temperature and salinity vertical profiles provided by the latest version of the World Ocean Atlas 2023 (WOA23), which integrates data from 1955 to 2022 at a resolution of $1°$ (Reagan et al., 2023). The WOA23 dataset is available at the NOAA website.

The sea ice concentration observational data used in this study is the EUMETSAT OSI-SAF (Lavergne et al., 2019). This

dataset is available at the Copernicus Climate Change Service Climate Data Store, and it provides coarse-resolution information based on measurements from various sensors, including the Scanning Multichannel Microwave Radiometer from 1979 to 1987, the Special Sensor Microwave/Imager from 1987 to 2006, and the Special Sensor Microwave Imager/Sounder from 2005 onward. The dataset covers the period from 1979 to the present day and is regularly updated. The grid resolution of this dataset is 25 km. In addition, we employ the Pan-Arctic Ice-Ocean Modeling and Assimilation System (PIOMAS), a coupled

model that integrates ocean and sea ice and assimilates daily satellite-derived products for sea ice concentration and sea surface

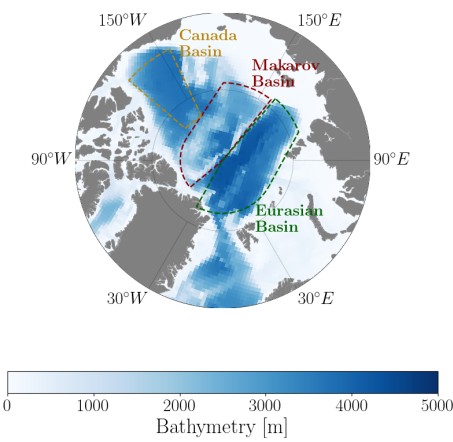

**Figure 2.** Bathymetry in meters of the ORCA1 configuration, derived from the ETOPO2 dataset. Dashed color lines show the boundaries of the Makarov (in red), Eurasian (in green) and Canada (in yellow) Basins (Peralta-Ferriz and Woodgate, 2015).

temperature (Zhang and Rothrock, 2003). The PIOMAS dataset covers the period from 1978 to the present, including Arctic sea ice thickness, which is utilized in this study.

## 3 Results

Our analysis focuses on the Arctic region, specifically the Makarov, Eurasian, and Canada Basins, which are characterized by
year-round sea ice coverage. These regions are defined as follows: The Makarov Basin (83.5–90°N between 50–180°W and 78–90°N between 141–180°E), the Eurasian Basin (82–90°N between 30–140°E and 78–82°N between 110–140°E), and the Canada Basin (72–84°N and 130–155°W)–see Fig. 2. Sea ice and upper ocean physical properties are closely linked due to mass and momentum exchanges at the ice-ocean boundary, exhibiting seasonal variations. In fall and winter, seawater freezes, and sea ice forms, accompanied by brine rejection. This process involves the rejection of salt from the crystal structure of
water ice, increasing salinity in the upper ocean layer. Consequently, ocean stratification weakens, leading to a deeper ML. In spring and summer, as sea ice melts, freshwater is released into the ocean, reducing salt concentration and strengthening ocean stratification, thereby causing the ML to become shallower.

### 3.1 Upper ocean properties

Fig. 3 illustrates the spatial distribution of mixed layer depth from control run and LOPS climatology in both March and
September. Model outputs have been averaged from January 1970 to December 2021, corresponding to the same years as the MLD from LOPS climatology. We observe that the MLD is generally underestimated by the model in September and March in the Arctic Basins, especially for the area of the Makarov and Eurasian Basins next to Greenland, where the model underestimates the MLD by tens of meters. Compared to a large portion of global models forced by CORE-II and JRA55-do, as studied by Ilıcak et al. (2016) and Allende et al. (2023), MLD discrepancies with observational data are less pronounced.



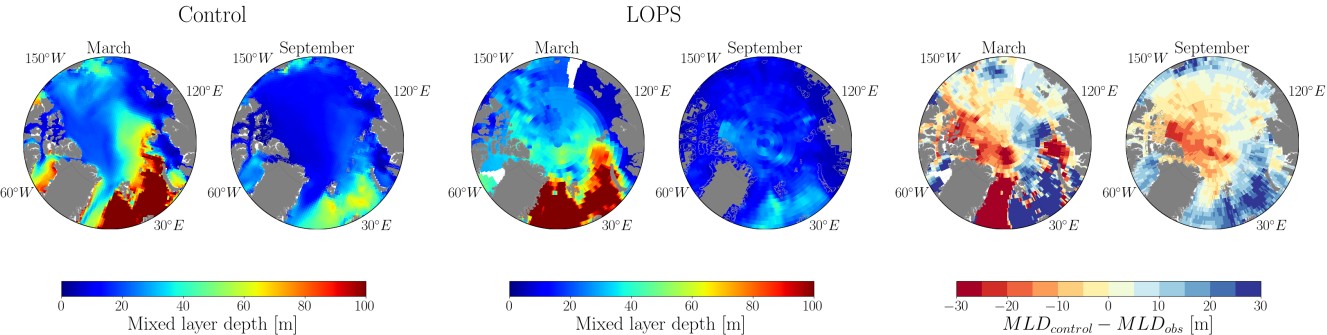

**Figure 3.** MLD maps of the LOPS climatology and the differences between NEMO-SI3 and LOPS climatology in March and September. Data is averaged in time between 1970-2021.

The area along the east coast of Greenland presents pronounced MLD differences in March, with shallower ML in the model than in the LOPS climatology. That region is covered with ice at that time of the year and no MLD observations exist there, while deeper ML are measured further offshore in the open and deep ocean. The climatology might result in an overestimation of MLD along that coast, relying on the only data present offshore for its mapping. This would be one point to be improved in a future version of this climatology. Furthermore, Fig. 4 displays the spatial distribution of sea surface temperature and salinity

for the control run and LOPS climatology. The sea surface temperature from the control run aligns with LOPS values for March and September in the three regions studied here. However, the March sea surface salinity from the control run appears to be saltier compared to the LOPS climatology over the Canada Basin and the eastern region of the Eurasian Basin, while it appears fresher than LOPS climatology in the region next to Greenland and the western part of the Makarov Basin in September.

We now investigate the sensitivity of the seasonal cycle of the MLD to the TKE MLP parameters previously introduced

in Section 2. Figure 5 illustrates the MLD seasonal cycle from the sensitivity experiments, the control run, and the LOPS climatology in each region. Data is averaged spatially for each basin. We observe similar behavior in all three regions when the TKE MLP parameters are varying. The most considerable difference between models is observed in the simulation with no attenuation to account for sea ice coverage (nn_eice=0). In such a case, a significant TKE MLP mixing is induced, resulting in a deeper ML for all months compared to other simulations. When using the other three options for the scaling parameter of the

TKE MLP (i.e., nn_eice=1, nn_eice=2, and nn_eice=3), the MLD values are closer to the LOPS climatology. We observe that both the control run (nn_eice=3) and the sensitivity experiment (with nn_eice=1) exhibit nearly identical seasonal cycles. This suggests that in these regions, the choice between attenuating mixing using a tangential hyperbolic shape or imposing a 25% of sea ice concentration limit for mixing yields similar outcomes. When the TKE penetration is turned off (rn_efr=0), the seasonal cycle is very weak and the MLD is underestimated in winter. Increasing the fraction of surface TKE that penetrates into the

ocean from 0 to 0.08 increases the MLD as well as the amplitude of the seasonal cycle. As expected, a weaker-amplitude seasonal cycle is observed when varying the type of exponential decay (nn_htau) from 30 to 10 m in high-latitude regions, as the TKE MLP vanishes more rapidly with depth.



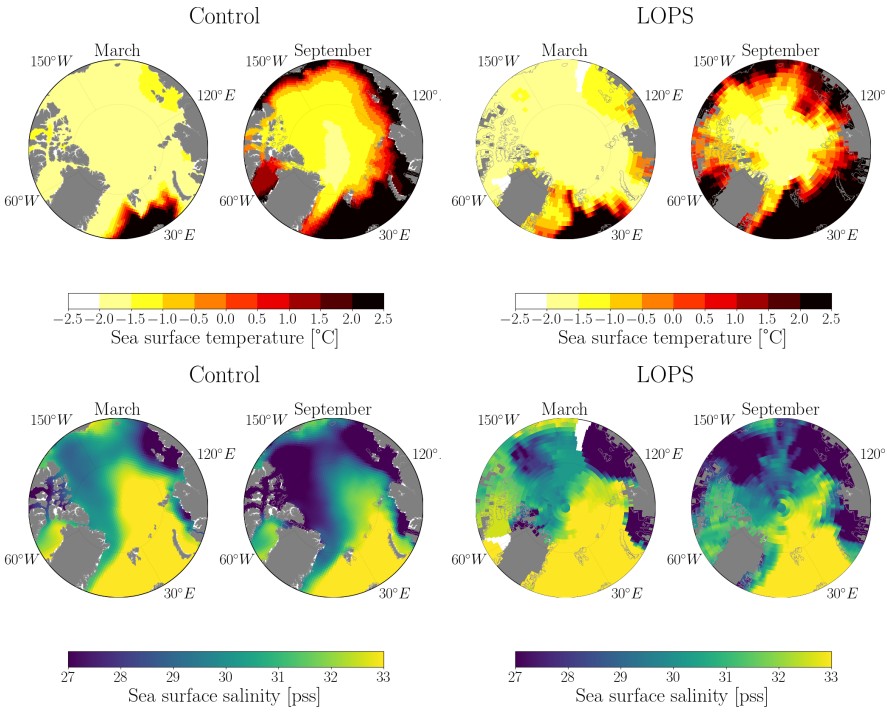

**Figure 4.** Maps of sea surface temperature and sea surface salinity for the Control run and LOPS climatology in March and September in the pan-Arctic region. Data is averaged in time between 1970-2021.

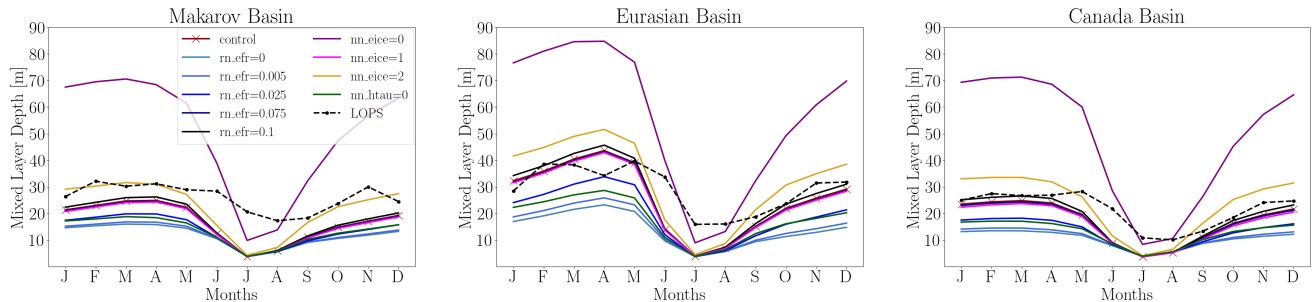

**Figure 5.** Seasonal cycle of the MLD in the Makarov, Eurasian and Canada Basins. Data is averaged in time between 1970-2021.

During the summer months of July and August, all simulations underestimate the MLD compared to LOPS climatology. These summer biases could be caused by the different reference depths used for computing the MLD with the density threshold criteria: 5 m for the LOPS climatology and 0.5 m for NEMO4.2. We recomputed the MLD using the 5m reference for the control run, and verified that it is not the case: the differences between the model MLDs computed with two reference depths were less than 5 m, and spatial patterns between the biases of observations and models were very similar (see Fig. A1 in the Appendix). The MLD std in summer is almost negligible, and in winter, for the Makarov and Canada Basins, it remains below



**Figure 6.** Maps displaying the differences in MLD between the sensitivity experiment (the title indicates the parameter modified) and the control run in March and September. Data is averaged in time between 1970-2021.

15 m, showing a similar spatial variability between experiments in these regions (see Fig. A2 in Appendix). However, for the

Eurasian Basin, differences between experiments appear to be more substantial. For instance, the MLD std reaches up to 30 m for the nn_eice=0 experiment and 8 m for the nn_efr=0 experiment.

A similar pattern emerges when examining the spatial distribution of the MLD. Fig 6 illustrates the differences between the sensitivity experiments and the control run for March and September. The largest ML deepening is observed for nn_eice=0 (no attenuation of mixing due to sea ice coverage), with MLD 20 m thicker than the control run in both months across the studied

regions. Similarly, nn_eice=2 (attenuation proportional to sea ice fraction) leads to a comparable pattern, with deeper ML ranging from 10 to 20 m in both months. Conversely, the most significant ML shallowing is observed for rn_efr=0 (TKE MLP turned off), resulting in ML shallower than the control by 20m, particularly in the Canada Basin. Decreasing the characteristic depth of TKE penetration from 30 m to 10 m (nn_htau=0) has an impact similar to a decrease of the penetrating fraction of energy rn_efr, with a similar spatial distribution in March. In the following, we will focus on three sensitivity experiments that



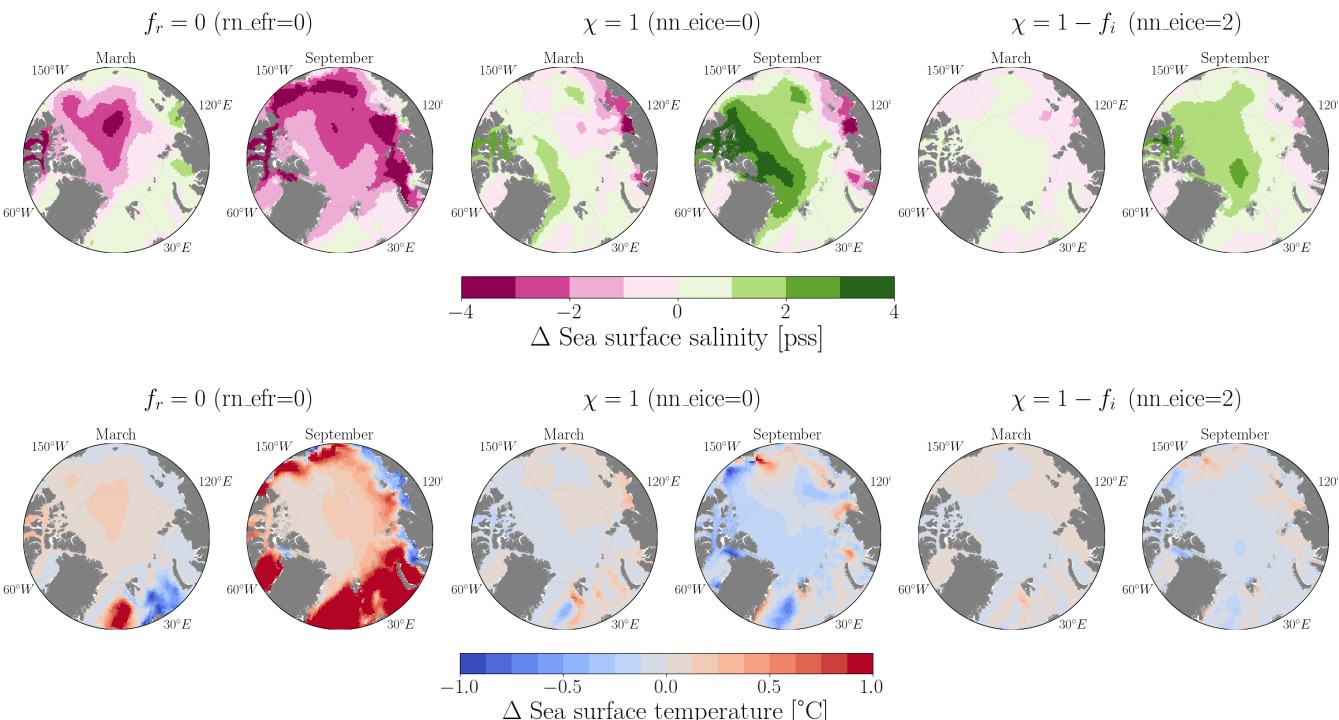

**Figure 7.** Sea surface salinity and temperature maps differences between sensitivity experiments and control simulation in March and September.

differ the most from the control simulation: rn_efr=0 (no TKE penetration) and nn_eice=0 (full TKE MLP under sea ice), as well as nn_eice=2 (default sea-ice dependency of the parameterization, used in CMIP6). Our analysis of the spatial distribution of sea surface salinity and sea surface temperature (see Fig.7) reveals that a decrease in the MLD corresponds to a reduction in sea surface salinity and an increase in the sea surface temperature compared to the control simulation. Conversely, an increase in MLD aligns with an increase in sea surface salinity and a decrease in sea surface temperature. A shallow ML results in less
mixing when the ice melts, leading to a fresh anomaly at the surface. In contrast, a deeper ML allows freshwater to mix deeper, resulting in higher salinity at the surface.

Looking into the vertical distribution of the ocean physical properties in September (Fig. 8), we observe that compared to the control run, the rn_efr=0 experiment shows an increase in upper ocean temperature, while the nn_eice=0 and the nn_eice=2 experiments exhibit a slight decrease, as previously observed in Fig. 7. The NEMO-SI3 simulations conducted here exhibit
similar behavior to the OMIP1 and OMIP2 simulations analyzed by Allende et al. (2023). For example, referring to Fig. 6 in the mentioned paper, we can observe that the salinity and temperature profiles of the IPSL model resemble those of the rn_efr=0 simulation (as expected, since they have deactivated the parameterization), while the CMCC profiles resemble those of the nn_eice=2 simulation. As studied by Ilıcak et al. (2016), the NEMO models Kiel-ORCA05, NOC, CMCC, and CERFACS underestimate the maximum temperature because the Atlantic water is not well simulated in this model group (refer to Fig. 7





**Figure 8.** Vertical temperature in $^\circ C$, salinity in pss, and Brunt-Väisälä frequency (N) in the Makarov, Eurasian, and Canada Basins (from left to right) in September. The shaded areas represent the variance. Data is averaged over the period from 1970 to 2021. The dashed lines represent WOA climatology.

of Ilıcak et al. (2016)). While our control simulation demonstrates improvements compared to these models, adjusting the TKE MLP parameters does not improve significantly the representation of the temperature maximum below 200 m. The biases with the WOA climatology for the maximum temperature are approximately 0.5$^\circ$C in all three basins. WOA climatology reaches values for maximum temperature of about 0.5, 1, and 0.5 $^\circ$C in the Makarov, Eurasian, and Canada Basins, respectively. Similar maximum temperature values are observed using the PHC3.0 climatology in the Eurasian and Canada Basins, as noted

by Ilıcak et al. (2016). Discrepancies between experiments are also observed in the vertical salinity profiles. Compared to the control run, the simulation with no TKE penetration (rn_efr=0 experiment) exhibits the freshest conditions, with a decrease in salinity of at least 2 pss in the first tens of meters across all three basins. At the surface, salinity increases as TKE MLP





intensifies; for example, in the nn_eice=0 experiment ($\chi = 1$), salinity increases by more than 2 pss compared to the control run across all three basins. The nn_eice=2 simulations, which utilize a scaling parameter of $\chi = 1 - f_i$, yield upper ocean salinity values similar to those from the WOA climatology. In March, upper salinity differences between the experiments and the control run decrease for nn_eice=0 and nn_eice=2 simulations in the three basins. However, they remain significant for the rn_efr=0 simulation (see Fig. A3 in Appendix for March profiles). Compared to the WOA climatology, except for the no TKE penetration simulation, all the others exhibit higher upper ocean salinity in the Makarov and Eurasian Basins. In the Canada Basin, upper ocean salinity values are similar to those of the control run simulation. To compare the stratification strength between the different simulations, we use the Brunt-Väisälä frequency, computed as

$$N = \sqrt{\frac{-g}{\rho} \frac{d\rho}{dz}} \tag{2}$$

Where g represents the acceleration due to gravity approximated to $9.8\,m/s^2$, $\rho$ is the potential density, and $z$ is the vertical distance measured upward. Large values of $N$ indicate a strong stratification and small values indicate a weak stratification. We observe a strong stratification in the control run and the rn_efr=0 experiment. On the contrary, the simulation with no attenuation of TKE MLP under sea ice is less stratified than the WOA climatology. This is in agreement with the MLD: with limited vertical mixing (rn_efr=0) a strong stratification is maintained in the upper layer, corresponding to shallow mixed layers. When TKE MLP is allowed under sea ice (nn_ice=0), the upper ocean is less stratified and the MLD is larger. This suggests that the scaling parameter $\chi$ which governs the TKE MLP under sea ice significantly influences the stratification. In the control run (nn_eice=3), TKE MLP vanishes as soon as the sea ice concentration reaches 25%, and there is not enough mixing (the stratification is too large compared to WOA climatology). This is improved when the TKE MLP varies proportionally to the sea ice concentration (nn_eice=2). These differences in ocean stratification have significant implications for freshwater content in the experiments. We compute the seasonal freshwater content (FWC) from May to December, using the May-average as a reference salinity, following the approach of Rosenblum et al. (2021). Discrepancies between experiments are evident in all sensitivity experiments compared to the control run. For nn_eice=0, FWC is notably over 2 m higher than the control run in the Canada Basin, where the FWC is approximately 5 m. At the same time, for rn_efr=0, it is less than 1 m (see Fig. A4 in the Appendix). However, the spatial patterns in the nn_eice=2 simulation are less pronounced; FWC slightly increases and decreases compared to the control run. These FWC results are consistent with those observed in the Community Earth System Models (CESM) study by Rosenblum et al. (2021).

## 3.2 Sea ice properties

Considering the impact of the TKE MLP parameterization on the upper ocean, we expect an impact on sea ice properties. Fig. 9 illustrates the spatial distribution of sea ice concentration and thickness from the control run and observational data in March and September. In March, the central Arctic Ocean is almost completely covered by sea ice, with concentration reaching nearly 100%, and sea ice thickness peaks between 2.5 and 3m. By September, the effects of summer melt become evident, leading to a noticeable reduction in both sea ice concentration and thickness. When comparing spatial patterns with observational data (OSI-SAF for sea ice concentration and PIOMAS for sea ice thickness), NEMO-SI3 exhibits lower sea ice concentration and



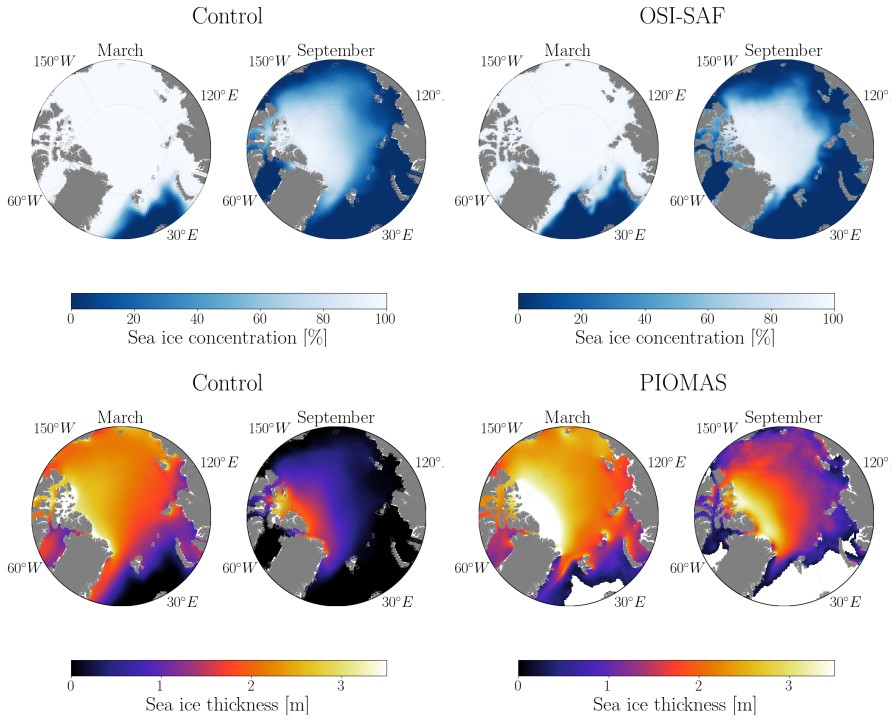

**Figure 9.** Maps of sea ice concentration and sea ice thickness for the control run and observational data in pan-Arctic regions during the months of March and September. The data has been averaged over the time period from 1970 to 2021 for the control run, and from 1979 to 2021 for observational data.

thickness in September. Specifically, regions near the East coast display sea ice thickness close to zero during this month. In March, biases between the simulated sea ice thickness and PIOMAS are more pronounced in the region next to Greenland. However, discrepancies between OSI-SAF and the simulated sea ice concentration seems to be relatively minor.

We analyze the seasonal cycle of sea ice concentration and thickness in the sensitivity experiments, control run, and ob-
270    servational data. Figure 10 illustrates these cycles across the Makarov, Eurasian, and Canada basins, with the model variables averaged from 1979 to 2021 to align with sea ice observational data. Compared with the OSI-SAF observational data, NEMO-SI3 performs well for sea ice concentration in winter months. However, during summer, the simulations underestimate sea ice concentration, with the most significant differences occurring in August, by approximately 30%, 48%, and 19% in Makarov, Eurasian, and Canada basins, respectively. These findings are consistent with similar observations made by Tsujino et al.
275    (2020) for OMIP models, by Wang et al. (2016) for CORE models, and for coupled models studied in the CMIP5 and CMIP6 exercises by Shen et al. (2021) in the Arctic region. Regarding sea ice thickness, NEMO-SI3 generally simulates thinner sea ice thickness in comparison to the PIOMAS reanalysis. Compared to the control run, these biases amount to approximately 70, 57, and 44 cm in March in the Makarov, Eurasian, and Canada basins, respectively. In September, the discrepancies reach 1.1 m across the three basins. Similar results were observed by Rosenblum et al. (2021) in the Canada Basin.



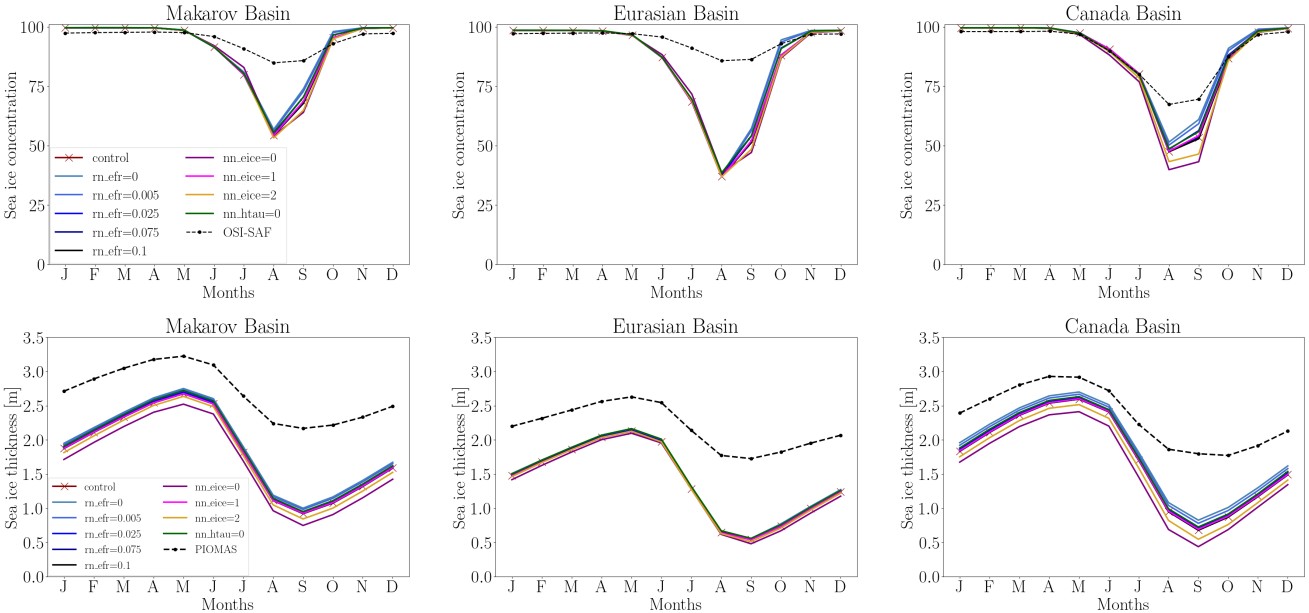

**Figure 10.** Seasonal cycle of the sea ice concentration and sea ice thickness in the Makarov, Eurasian and Canada Basins. Data is averaged in time between 1979–2021.

Differences between the experiments are relatively small, except in the Canadian basin in summer. The simulation with no TKE MLP (rn_efr=0) displays a 7% increase in sea ice concentration relative to the control case, while simulations with more TKE MLP under sea ice (nn_eice=0 and nn_eice=2) show a 10% and 7% decrease in September, respectively. A similar behavior is observed for sea ice thickness, with the largest differences between experiments observed in the Canada Basin. In September, in the rn_efr=0 experiment, sea ice thickness increases by 14 cm; in the nn_eice=0 experiment, sea ice thickness decreases by 24 cm; and the nn_eice=0 experiment shows a decrease of 13 cm in the sea ice thickness, relative to the control case. Fig. 11 shows the differences between the sensitivity experiments rn_efr=0, nn_eice=0, and nn_eice=2 with the control simulation. Notably, we observe similar spatial patterns between sea ice thickness and sea ice concentration: an increase in sea ice for the rn_efr=0 simulation and a decrease for the nn_ice=0 and nn_eice=2 simulations. For the nn_eice=0 simulation, a substantial reduction in sea ice—about more than 30 cm for sea ice thickness and 20 to 30% for sea ice concentration—is noticed in the area over the Beaufort gyre. While for the nn_eice=2 simulation, the spatial differences reach more than 10 cm. These discrepancies between experiments arise from variations in the density, salinity, and temperature vertical profiles (see Fig. 8). A strong stratification leads to a large Richardson number, which restricts vertical mixing and its associated upward vertical heat flux. This reduction in vertical heat fluxes implies less exchange between the upper ocean and sea ice, resulting in reduced sea ice melt during the summer months, as observed in the experiment without TKE MLP (nn_eice=0). In contrast, a weak stratification enhances vertical mixing and heat flux, leading to increased sea ice melt, as seen in experiments with more TKE MLP under sea ice (nn_eice=0 and nn_eice=2).



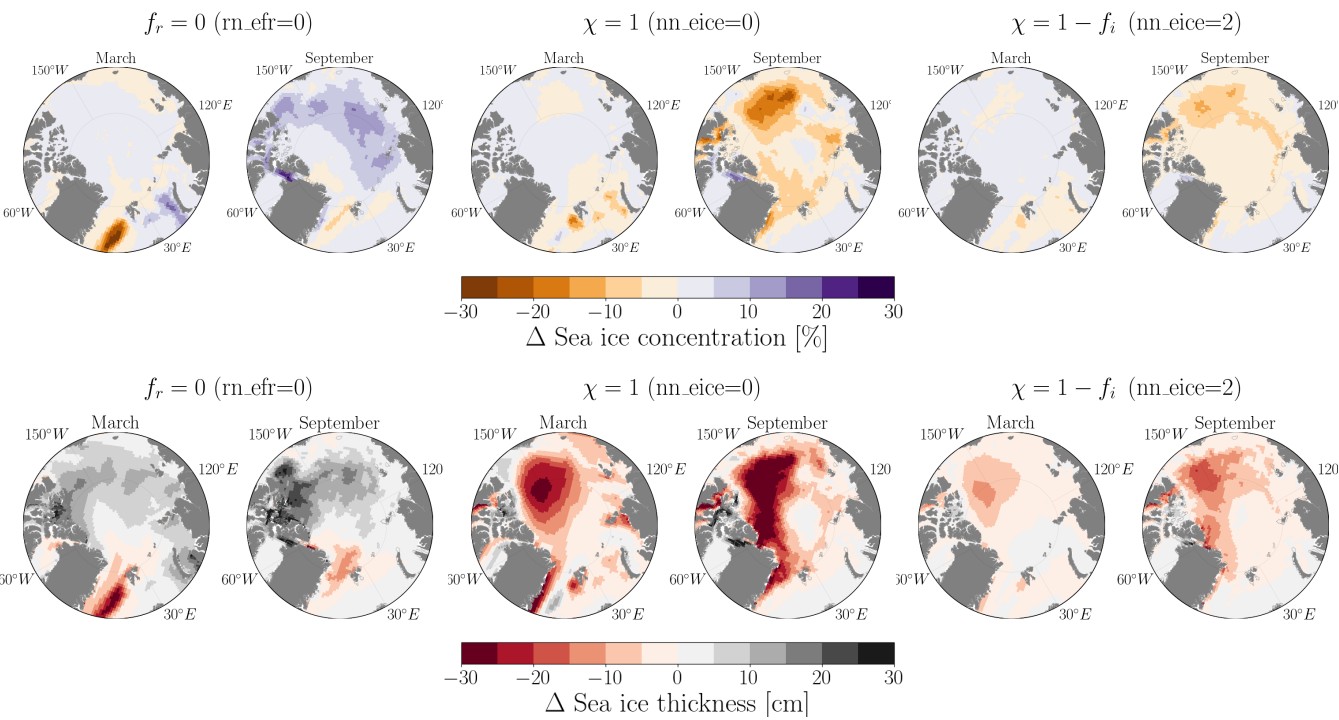

**Figure 11.** Maps of sea ice concentration and thickness differences between the sensitivity experiments and the control run in March and September. Data is averaged over the period from 1970 to 2021.

## 3.3 Inter-annual variability

We now examine the summer and winter inter-annual variability of the upper ocean and sea ice properties in the control run and three sensitivity experiments: rn_efr=0, nn_eice=0, and nn_eice=2. Fig. 12 displays the evolution of the MLD, sea ice

concentration and sea ice thickness during the summer months of June to September. Previous studies by Cole and Stadler (2019) and Wei et al. (2024), utilizing ITP observational data, highlight the increase in the MLD in the Canada Basin since 2000. We also observe this trend in the MLD from ITP observational data in the three basins, where the linear regression slopes are 0.54, 0.37, and 0.19 m/year for the Makarov, Eurasian, and Canada Basins. It is important to note that while this trend is well-captured in the Canada Basin due to extensive ITP observations, the coverage in the Makarov and Eurasian Basins is

limited (see Fig. A6 in Appendix). Although we observe an increase in the MLD in these regions, further data collection is necessary to fully assess the reliability of this trend.

Consistent with Fig. 5, all experiments simulate a ML shallower than observations, except for the nn_eice=0 experiment with strong TKE MLP under sea ice, which exhibits an unrealistic seasonal cycle. Specifically, it underestimates the MLD in July and August but overestimates it in June and September. We compute the same regression for sensitivity experiments

within the same time frame (from 2000 to 2021). The rn_efr=0 experiment (no TKE MLP) does not show an increase in the



**Figure 12.** MLD, sea ice thickness and sea ice concentration for each summer (June to September) from 1970 to 2021 in the Makarov, Eurasian, and Canada Basins. Solid lines represent the linear regression, and $m$ denotes the slope.

MLD, displaying a nearly flat linear regression. All other experiments display increasing trends in the three basins, except for the nn_eice=0 experiment, which shows a slight decrease in the Eurasian Basin (m=-0.15 m/year).

We also compare the summer inter-annual variability of the simulated sea ice concentration and thickness with OSI-SAF observational data and PIOMAS reanalysis, respectively. Regarding sea ice concentration, significant biases are evident in the
Eurasian Basin, whereas the simulated sea ice concentration aligns more closely with observational data in the Makarov and Canada basins. The experiments also underestimate the sea ice thickness compared to the PIOMAS data in the Makarov and Eurasian Basins, although the biases seem lower after year 2000. In the Canada Basin, the sea ice thickness is very close to the PIOMAS reanalysis during the full period. Focusing on the trend from 2000 to 2021, both observational data and model simulations show a decrease in sea ice concentration in all three basins. The decrease is stronger in the model in the Makarov
and Eurasian Basins compared with observations; for instance, the slopes of the linear regression reach -1.18 %/year for the control run and -0.32 %/year for the OSI-SAF data in the Eurasian Basin. However, in the Canada Basin, the NEMO-SI3





**Figure 13.** MLD, sea ice thickness and sea ice concentration for each winter (October to April) from 1970 to 2021 in the Makarov, Eurasian, and Canada Basins. Solid lines represent the linear regression, and $m$ denotes the slope.

captures the decreasing trend in sea ice concentration quite well, with similar slopes for the sensitivity experiment linear regressions compared to the OSI-SAF one. This declining trend has been previously observed and studied by Tsujino et al. (2020), Stroeve and Notz (2018), and Cavalieri and Parkinson (2012) from both observational data and model simulations. This

short-term trend is also evident in the simulated sea ice thickness and the sea ice thickness from the PIOMAS reanalysis in all three basins. The loss of sea ice thickness appears to be slower than that of sea ice concentration, with a slope of approximately -0.04 m/year for the PIOMAS reanalysis in all the basins.

Fig. 13 shows the winter inter-annual variability of the MLD, sea ice concentration, and sea ice thickness computed as the monthly average from October to April. As was the case in summer, the inter-model comparison reveals substantial differences

in MLD. The overestimation of the MLD when TKE MLP is kept under sea ice (nn_eice=0), also shown in Fig. 5, as well as the underestimation when there is no TKE MLP (rn_efr=0) are very clear. Comparing simulations with ITP observational data, the nn_eice=2 and control simulations exhibit a closer resemblance. It has already been observed for the MLD seasonal cycle





analysis. The increasing trend in MLD since 2000 is observed in ITP observational data in the Makarov and Canada Basins, with values for the linear regression slope being 0.88 and 0.93 m/year, respectively. Once again, the nn_eice=2 and control

simulations closely resemble this behavior.

The winter inter-annual sea ice concentration reaches values close to 100%, dropping to around 98% in some years over the Canada Basin for all simulations except the one with vanished TKE MLP; those values are very similar to OSI-SAF observational data, except at the beginning of 1980 where the sea ice concentration decreases until 94%. Compared with PIOMAS reanalysis, larger sea ice thickness values are noted in the Makarov and Eurasian basins than those simulated. Simulated sea

ice thickness ranges from 1.5 to 2 m, exhibiting a declining trend since 2000, consistent with trends observed in the PIOMAS reanalysis.

The decline of sea ice concentration since 2000 could explain the increase in the MLD for the nn_eice=2 and control run simulations (nn_eice=3) because both parameterizations involve a scaling parameter depending on sea ice concentration. This is not the case for the other two cases, nn_eice=0, where TKE MLD is present everywhere, and rn_efr=0, where TKE MLP

is not activated. Indeed, when we look at the summer and winter inter-annual variability in the Canada Basin for the full set of sensitivity experiments (see Fig. A5 in Appendix), we observe that the MLD increase is present for the large part of the experiments, except the full TKE MLP (nn_eice=0) and those having a small percentage of the surface TKE penetrating in the ocean (rn_efr=0 and rn_efr=0.005).

## 4   Discussion and conclusion

We analyzed the NEMO-SI3 model's response to changes in the TKE MLP scheme within the central Arctic Ocean, focusing specifically on the Makarov, Eurasian, and Canada basins. This ad-hoc parameterization adds to the standard TKE parameterization an extra source of mixing that represents the effect of near-inertial oscillations and ocean swells affecting below the mixed layer (ML). This parameterization is governed by three key parameters: $f_r$, which denotes the fraction of surface TKE that penetrates into the ocean; $h_\tau$, representing the vertical mixing length scale that controls the exponential shape of the

penetration; and the scaling parameter $\chi$ in response to the presence of sea ice. Our investigation identified three contrasted settings with significant implications for ocean and sea ice dynamics: when the TKE MLP parameterization is deactivated ($f_r = 0$, referred to as the rn_efr=0 experiment); when the TKE MLP is maximum due to the absence of attenuation by sea ice ($\chi = 1$, labeled as the nn_eice=0 experiment); and when sea ice attenuation is proportional to its concentration ($\chi = 1 - f_i$, designated as the nn_eice=2 experiment). Compared to the control simulation ($f_r = 0.08, \chi = 1 - \min(1, 4f_i)$, and $h_\tau = 30m$),

the winter MLD was exceptionally large for the full TKE MLP mixing case (nn_eice=0), moderately increased for nn_eice=2, and decreased for rn_efr=0. Comparing these simulations with observational data revealed that extreme cases, the full TKE MLP mixing and the no TKE penetration, are unrealistic. As noted by Calvert and Siddorn (2013), Rodgers et al. (2014) and Storkey et al. (2018), the additional source of mixing by TKE MLP is beneficial in the NEMO model. Our extreme experiment nn_eice=0 demonstrates that this source of mixing needs to be attenuated in the presence of sea ice. This is obvious from a

physical point of view, because sea ice isolates the ocean from the atmosphere and damps inertial oscillations (Rainville et al.,





2011). In this study, we have compared different functional forms for this attenuation. $\chi = 1 - f_i$ is the default option in NEMO, used in most CMIP6 projections, which displays a good agreement for the seasonal cycle of the MLD in the Makarov Basin. However, it results in a stronger seasonal cycle of the MLD compared with observations over the Eurasian and Canada Basins. The two other options $\chi = 1 - \tanh(10f_i)$ (nn_eice=1) and $\chi = 1 - \min(1, 4f_i)$ (our control case) behave similarly and pro-

duce a seasonal cycle of MLD closer to the LOPS climatology in these regions. Nevertheless, during summer all experiments underestimate the MLD over the three basins compared to LOPS climatology.

We assessed the variations in upper vertical properties, revealing surprising discrepancies in ocean stratification. The experiment with full TKE MLP mixing (nn_eice=0) displayed minimal stratification. In contrast, the experiment with vanished TKE MLP mixing (rn_efr=0) exhibited a strong stratification, similar to the control run. The nn_eice=2 experiment displayed

stratification between both extreme experiments. This suggests that the choice of the scaling parameter significantly affects ocean stratification and the density profile over the first 30 m of depth. Moreover, we observed that changes in stratification are consistent with the changes in MLD: stronger stratification corresponds to shallower MLD, while weaker stratification corresponds to deeper MLD. These changes imply significant differences in salinity near the surface. We observed an increase in salinity for simulations with an increased MLD. Conversely, simulations with decreased MLD exhibited a decrease in sea

surface salinity. Furthermore, we examined the seasonal freshwater content, which is closely linked to discrepancies in ocean stratification. For the nn_eice=0 simulation, the FWC measures more than 2 m higher than the control simulation near the Canada Basin, reaching more or less 5 m values. For the rn_efr=0 simulation, the FWC decreased by about 1.5 m in the three basins.

We have investigated for the first time whether the TKE MLP scheme influences sea ice concentration and thickness. Overall,

the underestimation of sea ice thickness and sea ice concentration in the control run remains in all sensitivity experiments, which show that these biases are not due to vertical mixing only. Nevertheless, the sea ice thickness is sensitive to vertical mixing. We found a reduction from 30 to 40 centimeters compared to the control case when mixing was large under sea ice (no TKE MLP attenuation) or not completely turned off (nn_eice=2). Conversely, the suppression of TKE MLP created a moderate increase in sea ice thickness, ranging from 10 to 20 centimeters. These discrepancies between experiments derive

from variations in the salinity and temperature vertical profiles. A strong stratification results in a large Richardson number, which restricts vertical mixing and its associated upward vertical heat flux. In contrast, a weak stratification enhances vertical mixing and heat flux.

Furthermore, we examined the inter-annual variability of the MLD and sea ice properties, yielding results consistent with our seasonal analysis. We observe a short-term trend of MLD from ITP observational data since 2000 in the three basins during

summer and winter. The summer MLD has increased at a rate of 0.54, 0.37, and 0.19 m/year in the Makarov, Eurasian, and Canada basins, respectively. The winter MLD has only increased in the Makarov and Canada Basins with rates of 0.88 and 0.93 m/years, while the Eurasian Basin decreased at -0.15 m/years. It is important to highlight that while this trend is well-captured in the Canada Basin due to extensive ITP observations, the coverage in the Makarov and Eurasian Basins is limited. Although we observe an increase in the MLD in all three regions, further data collection is necessary to assess whether the ML

is deepening not only in the Canada Basin but also in the Makarov and Eurasian Basins. The simulations with no TKE MLP do





not represent the MLD trends. This raises questions regarding the choice made in some CMIP6 models to remove TKE MLP, which may impact the representation of future arctic trends.

Our study highlights the significant impact of the TKE MLP parameterization on upper ocean and sea ice properties in the Arctic Ocean. This parameterization has been widely utilized in the NEMO community. It aims to reproduce the upper
ocean vertical mixing driven by near-inertial oscillations, ocean swells, and waves. However, this parameterization lack a physical basis and is considered "ad hoc" (Rodgers et al., 2014). Furthermore, the TKE MLP scheme in NEMO appears to be time-step dependent, yielding different results as the resolution changes. More generally, the complexity of the various TKE options has led some groups to use the generalized length scale approach instead of the "historical" TKE vertical mixing scheme (Reffray et al., 2015). Another alternative vertical mixing scheme has been developed as part of the UK OSMOSIS
project, planned to be included in the UK Global Ocean GO8 (Storkey et al., 2018). This initiative aims to refine near-surface oceanic mixing characterization through a combination of observational campaigns and a novel mixing scheme derived from extensive large eddy simulations. Future research should deepen our understanding of the underlying mechanisms driving the TKE MLP parameterization and explore alternative approaches to improve the robustness and accuracy of vertical mixing parameterizations in NEMO, especially in the presence of sea ice. Such efforts will be crucial for enhancing the fidelity of
Arctic climate projections and advancing our understanding of polar climate dynamics.

*Code and data availability.*  The NEMO version 4.2.1 utilized in this study is accessible in Zenodo via Allende (2024). For detailed information on version 4.2.1 utilized in this study, refer to the official user guide. For additional details regarding the latest version of the NEMO code, please refer to the NEMO's official repository. Model outputs, simulation details, and Python scripts to reproduce our figures are available in the supplementary materials provided by Allende Contador (2024).

**Appendix A**

**A1**

We here compile additional figures related to: MLD maps illustrating the differences between MLD from different reference depth criteria (Fig. A1); standard deviation seasonal cycle of the MLD in the Makarov, Eurasian and Canada Basin (Fig. A2); vertical temperature, salinity and Brunt-Väisälä frequency in the Makarov, Eurasian, and Canada Basins in March (Fig. A3);
spatial distribution of the differences in the seasonal freshwater content between the experiments and control runs from May to December (Fig. A4); MLD inter-annual variability in summer and winter in the Canada Basin (Fig. A5); and MLD map from ITP data over the Arctic Ocean (Fig. A6).



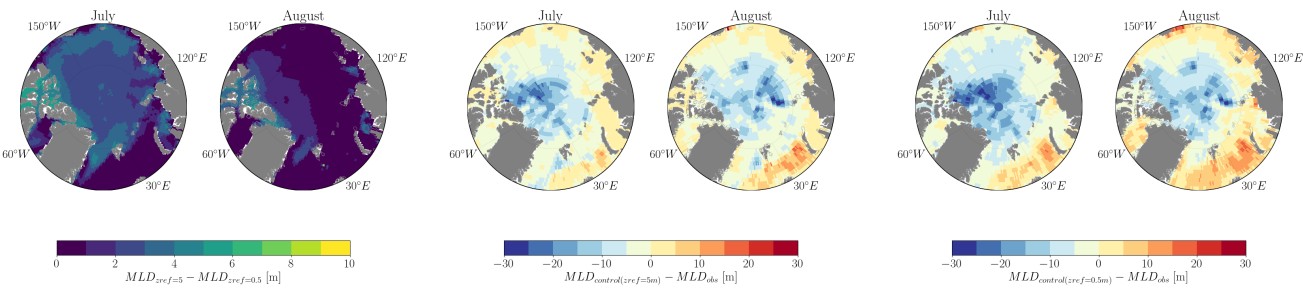

**Figure A1.** MLD maps illustrating the differences between the MLD from the control run computed at $z_{ref} = 5$m and $z_{ref} = 0.5$m, as well as the corresponding differences with the LOPS climatology during July and August.

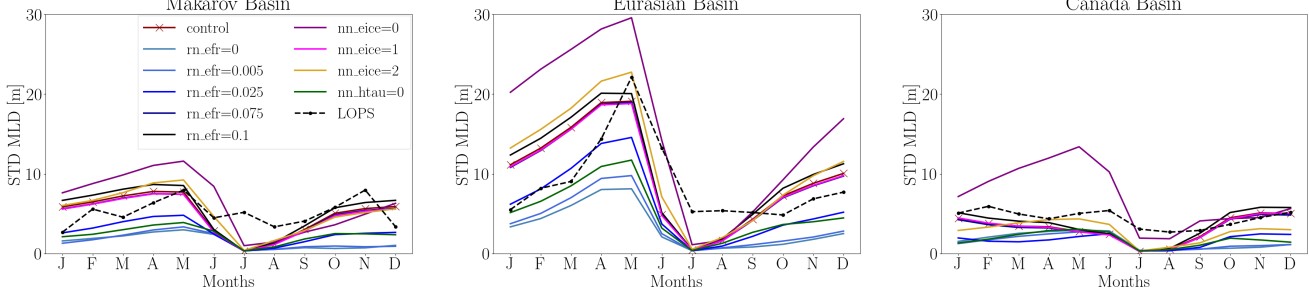

**Figure A2.** Standard deviation seasonal cycle of the MLD in the Makarov, Eurasian and Canada Basin. Data is averaged in time between 1970-2021.



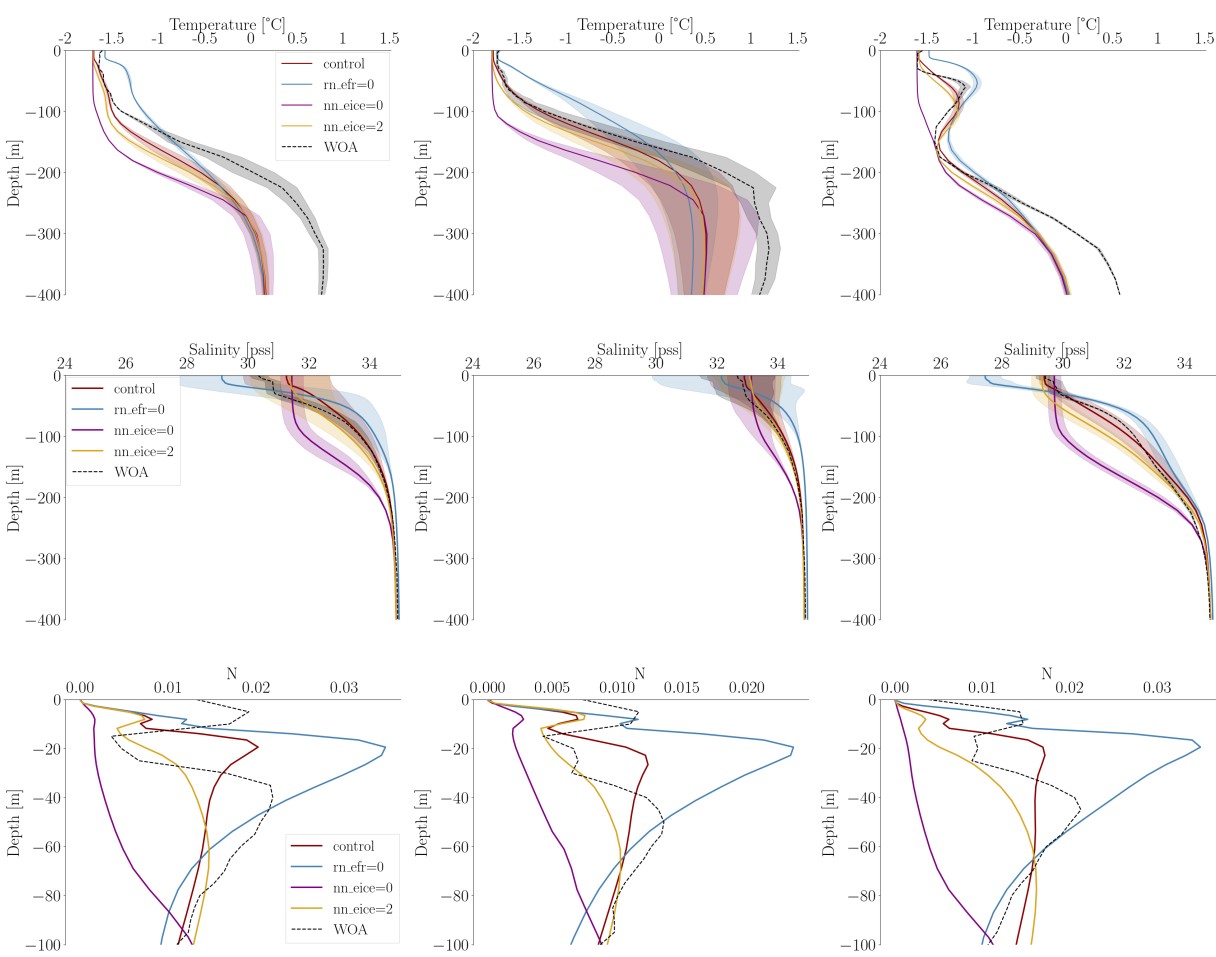

**Figure A3.** Vertical temperature in $°C$, salinity in pss, and Brunt-Väisälä frequency (N) in the Makarov, Eurasian, and Canada Basins (from left to right) in March. The shaded areas represent the variance. Data is averaged over the period from 1970 to 2021. The dashed lines represent WOA climatology.





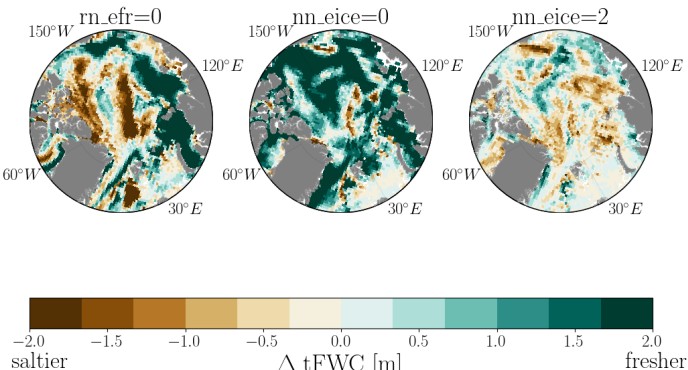

**Figure A4.** Maps of the differences in seasonal freshwater content (FWC) between the experimental and control runs from May to December. Data spans from 1970 to 2021.

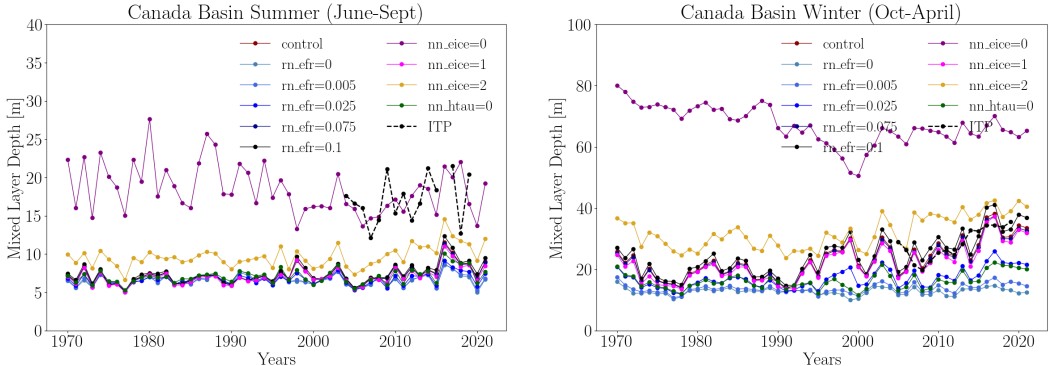

**Figure A5.** MLD inter-annual variability in summer (June to September) and winter (October to April) from 1970 to 2021 in the Canada Basin.

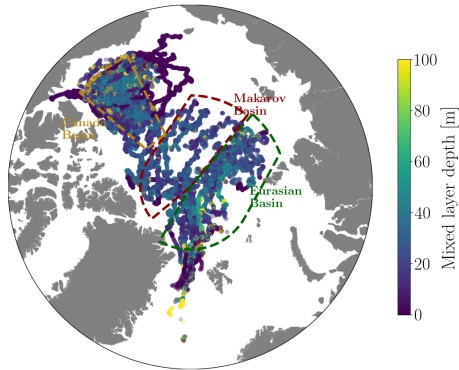

**Figure A6.** MLD map from ITP data over the Arctic Ocean. Dashed-lines represent the boundaries of the Makarov Basin, Eurasian Basin, and Canadian Basins.





*Author contributions.* SA, AMT, and CL collectively contributed to conceptualizing the research outlined in this paper. SA conducted the simulations, performed the statistical analyses, and created all the figures. SA took the lead in writing the manuscript, with contributions
from AMT, CL, CDB, FM, TF, and AB.

*Competing interests.* The authors declare that they have no known competing financial interests or personal relationships that could have appeared to influence the work reported in this paper.

*Acknowledgements.* This work was conducted within the JPI Oceans and Climate project MEDLEY (MixED LayEr heterogeneitY), which is partly funded by the Belgian Science Policy Office under contract BE/20E/P1/MEDLEY; and by the French Agence Nationale pour la
Recherche under contract 19-JPOC-0001-01.

Computational resources have been provided by the supercomputing facilities of the Université catholique de Louvain (CISM/UCL) and the Consortium des Équipements de Calcul Intensif en Fédération Wallonie Bruxelles (CÉCI) funded by the Fond de la Recherche Scientifique de Belgique (F.R.S.-FNRS) under convention 2.5020.11 and by the Walloon Region



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
