# Peer review of "Impact of ocean vertical mixing parameterization on Arctic sea ice and upper ocean properties using the NEMO-SI3 model"

_Geoscientific Model Development, 2024_

## Referee Comment (RC2)

Review of "Impact of ocean vertical mixing parameterization on Arctic sea ice and upper ocean properties using the NEMO-SI3 model" by Allende et al., (gmd-2024-49)

This studies tests a general modification to turbulence parameterization that redistributes turbulent kinetic energy across the base of the mixed layer to balance existing underestimations of the mixed layer depth, with respect to its performance on surface ocean and sea ice properties in the Arctic by evaluating multi-year runs with different parameter settings. Turbulence parameterizations are an important and critically underconstrained factor in ocean and climate models, and their improvement is vital to reduce uncertainties in climate predictions. The methods are established and sound, and the paper will be a good and important contribution to the improvement of ocean models.

Prior to publication, I believe the following major concerns would need to be addressed:

(1) Throughout the manuscript, the authors use both the assigned variables (chi, f_r, h_T) as well as the NEMO-internal identifiers (rn_efr, nn_htau etc) to refer to the paramters. This makes the manuscript very hard to follow, and gives the impression of a NEMO-specific technical report rather than a scientific paper. I strongly recommend removing all NEMO-internal identifiers and only use the actual variables. (If needed, a table could be added in the Appendix to relate to the NEMO identifiers, but I assume these can be found in a NEMO-specific documentation. I would also try to reduce the references to NEMO specific models to a minimum, and mention other model frameworks that use TKE MLP to make the study more appealing for readers outside the NEMO community

(2) In the results section, surface ocean properties are discussed individually, but these values are directly linked in a very straightforward and known way (more mixing = redistribution of salt = deeper ML = higher surface salinity = lower stratification). This separation leads to repetition and makes it hard to follow the paper. I recommend re-arranging both results sections to present findings on the individual diagnostics (MLD, surface salinity, stratification) side by side.

(3) It would be great to have a table summarizing which configuration performs good/intermediate/bad for specific regions (the three basins) and processes (seasonal cycle, interannual trends, MLD; surface salinity, sea ice).

(4) The overwhelming part of the discussion and conclusion section is a repetition of the results. The summary should be massively shortened, to one paragraph, and the discussion should address questions like: Why do the different configurations yield the observed difference in simulated conditions? What is your recommendation for the 'best' configuration of the TKE MLP in Arctic Environments? How much better does your identified best choice perform against the state-of-the-art configuration? Is it generally applicable, or do users need to discriminate which configuration to use based on regions? What are the limitations of the parameterization, ie, where does it not perform well? What data is needed to make further improvements? Is this parameterization the way to go ahead, or are there alternatives which might be superior in the long run?

Minor concerns:
- The description of the implemented underlying turbulence parameterization (not MLP) used in the methods section (l. 60-71) is very brief. For a manuscript focussing on improving turbulence parameterization, I would appreciate a more detailed description without having the reader go back to the original literature.
- l. 95 I do not understand what "excluding salinity restoration under sea ice" means, please add a brief explanation.
- l. 96 Rathore et al. - year is missing
- Table 1 is not necessary, is / can be incorporated in the text
- Ocean and Sea Ice variables around l. 115: No need to give the NEMO-internal identifiers, stick to the commonly used symbols.
- l. 125. The density threshold criterion is repeated in two different sentences
- l. 127 "The choice of the reference depth is impactful" - please provide details on why it is, and also I do not understand why you pick different reference depths for model / obs later on, what was the reason for that (especially as you state the choice does NOT make a difference, around l. 195)
- if nighttime convection is not an important issue in the Arctic, shorten the corresponding paragraph
- Table 2. Needs improvement, I do not understand it.
- l. 137. A citation is needed to the TEOS-10 part, no need to mention a python toolbox was used. Also, consistently using TEOS-10 would make it consequent to use absolute salinity and not practical salinity (which is unitless, not pss) throughout the manuscript. The TEOS-10 framework also includes the Brunt-Väisäila frequency $N$, which should be introduced here instead of around l. 240 in the results, with a shorter description.
- l. 157-162: you introduce upper ocean and sea ice variability by describing the seasonal cycle, but there are also huge regional differences in the Arctic Ocean that should be summarized here.
- l. 169: "MLD discrepancies are less pronounced" compared to other NEMO simulations, or any other simulations in general?
- l. 173: "This would be one point to be improved in a future version of this data set" belongs into the discussion.
- l. 195 (and around): Confusing, why not just use the same reference density? Also, move to methods.
- I am not sure if I understand the meaning of the standard deviation comparison: Is that a measure for the internal variability in the three basins? Please expand.
- l. 214-216: Not only sea ice melting affects salinity anomaly, also advection of (modified) river water!
- l. 219-230: This comparison to other model runs feels out of place here, but could be modified and go into the discussion. Also, Atlantic Water temperature at 200m depth should be relatively unaffected by modifications of the surface(!) mixing parameterization, right?
- Figures 7, 8: specify which temperature is shown (potential, conservative)?
- l. 239ff. Higher turbulence reduces stratification is quite basic - I am not sure if this paragraph adds to the story of the paper. Consider to remove?

- l. 251- l. 257. I cannot follow the FWC part: Mixing only redistributes salt / FW, so is the difference from additional ice melt? The numbers seem high for that. Also, for FWC results, a discussion of the sensitivity to choice of reference salinity is needed.
- l. 260: Add HOW sea surface properties affect sea ice.
- l. 266 what is meant by "East coast"
- l. 292 (and again later): remove "leads to a large Richardson number, which" - the Richardson number has never been defined here, relation stratification and Ri is trivial, so it adds no information here.
- l. 308. "unrealistic seasonal cycle" - unrealistic in which way?
- remove "displaying a nearly flat linear regression", that is the same as 'no trend'
- l. 363. "beneficial in the NEMO model" - in which way and where? Open water?
- l. 386 "biases are not due to vertical mixing" - what could be likely other reasons for these biases then?

---

## Author Comment (AC1)

July 1, 2024

Sofía Allende
ELIC Place Louis Pasteur 3/L4.03.08
1348 Louvain-la-Neuve Belgique
Phone: +(33) 641842014
Email: sofia.allende@uclouvain.be

This is a useful modelling study with a finding that differences in ocean stratification due to TKE MLP parameters have significant implications for the Arctic freshwater content. For global NEMO modellers would be nice to know how these various simulations perform in other sea-ice covered seas, not just in the Arctic. Also, is there a specific configuration that is recommended and differs from default settings?

The text could be improved significantly. Its expressions are often confusing and the writing style could be clearer, in particular scientific terms and simulation names could be used in a more uniform manner, which would help reading a lot. I give some detailed comments to assist on this:

We thank the Referee for his/her detailed assessment of our manuscript and suggestions aimed at improving the clarity and relevance of our work. We agree that it would be valuable to understand how our various simulations perform in sea-ice covered regions beyond the Arctic, and it is one of the perspectives of this work as for instance evaluate them on the Southern Ocean, the Sea of Okhotsk, and the Bering Sea. However, we decided to focus on the Arctic region because there are more observational data available for validation. We have added this point to the conclusion and discussion section.

Additionally, we have revised the conclusion and discussion section to include recommendations for the TKE MLP in the regions studied here. We have integrated a summarized table detailing the performance of each simulation across the seasonal cycle and inter-annual variability of the MLD and sea ice thickness in each region. We have also provided a discussion on the limitations and potential improvements of the scaling parameter.

Hereafter, we give a short list of the implemented changes in view of the Referee's comments. Please find below a part-by-part reply.

- line 58: Instead of writing 'several regions', be specific and write '3 regions'
Included (line 69). We changed the phrase to: "... the upper ocean and sea ice properties in three regions of the Arctic Ocean ..."

-line 58: 'by varying the mixing scheme' is more clear, if that is meant here. 'varying the mixing just below the mixed layer TKE parameters' is hard to understand.
Included (line 69). We replaced it with "by varying the mixing scheme".

-line 61: Change to 'The vertical turbulent kinetic energy (TKE) closure scheme'
Done (line 72). We changed it to: "The vertical turbulent kinetic energy (TKE) closure scheme implemented in NEMO is based on the model developed by Bougeault et al. 1989 for the atmospheric boundary layer."

-line 63: Change to 'integrated by Blanke and Delecluse (1993) into the OPA model'.
Included.

-line 75: 'ocean's surface boundary layer'.

Done.

-line 84: Can f_r be > 0.1? If it can, would it make sense to try higher values?
We thank the Referee for this question. The upper limit for the f_r parameter is 0.1, as it is set in the NEMO model. Therefore, it is not possible to evaluate higher values.

-line 84: Do you mean that 5% of TKE is redistributed below MLD or in the MLD and below it?
We thank the Referee for pointing out this misleading phrasing. It means that 5% of TKE is redistributed within the MLD and below it. We have significantly revised this section of the text and removed this sentence for the sake of clarity.

-line 94: Explain what ORCA1 is.
We have included the following: "To carry out this investigation, we utilize the NEMO4.2 version with the SI3 sea ice model, using the eORCA1 configuration. The eORCA1 quasi-isotropic global tripolar grid has a nominal resolution of $1°$, extended to the south to better represent the contribution of Antarctic under-ice shelf seas to the Southern Ocean freshwater cycle. The grid has a latitudinal grid refinement of $1/3°$ in the equatorial region. The vertical discretization consists of 75 levels, where the initial layer thicknesses increase non-uniformly from 1 m at the surface to 10 m at 100 m depth, reaching 200 m at the bottom." (lines 103-107).

-line 97: 'Rathore et al. (2024)' [year is missing]
We added the year to the reference.

-Table 1 and others: Table captions are usually above tables.
We have modified the position of the captions in the tables.

-line 114: Is pss Practical Salinity Scale? Does it equal psu which is more commonly used?
We thank the Referee for this question. In NEMO the salinity unit used for the equation of state is the practical salinity scale.

-line 120: 'de Boyer Montegut (2024)' [typos in the reference]
We corrected the reference to: "de Boyer Montégut C.: Mixed layer depth over the global ocean: a climatology computed with a density threshold criterion of 0.03 kg/m3 from the value at the reference depth of 5 m, https://doi.org/10.17882/98226, 2024."

-line 125: 0.03 kg/m$^3$ [there should be space between number and unit, change also line 127]
Done.

-Table 2: It is difficult to follow in text what simulation is what. Perhaps number simulations in this table from 1 to 9 and use those numbers consistently in text when refererring to particular simulations.
We thank the Referee for the suggestion. Following the comments of Referee 2, we have modified the text to refer to the simulations using the names of the parameters in the equation ($f_r$, $\chi$ and $h_\tau$), rather than the individual names of the NEMO parameters (rn_efr, nn_eice and nn_htau).

-line 137: 'We computed'
We added this modification, the new phrase is (lines 149-151): "We computed vertical potential density profiles using the TEOS-10 Gibbs Sea Water toolbox (Mcdougall et al (2011), and determined the MLD by applying the threshold density criteria. This was done to compare ITP observational data with NEMO sensitivity experiments, where the surface reference depth for ITP varies from 10 to 0 m depending on the profile."

-line 140: 'The ITP data are from'
Included (line 152). The new text is the following: "The ITP data are from 2004 until 2019, with the majority of the observations between the years 2007 to 2015. "

-Figure 2 caption: 'Basins following Peralta-Ferriz and Woodgate (2015).
We have corrected this caption to: "Figure 2. Bathymetry in meters of the ORCA1 configuration, derived from the ETOPO2 dataset. Dashed color lines show the boundaries of the Makarov (in red), Eurasian (in green) and Canada (in yellow) Basins following Peralta-Ferriz and Woodgate (2015)."

We thank the Referee for pointing out this mistake. Effectively, the boundaries for the Eurasian Basin were incorrect. The text has been revised as follows: "Our analysis focuses on the Arctic region, specifically the Makarov, Eurasian, and Canada Basins, which are characterized by year-round sea ice coverage. These regions are defined as follows: The Makarov Basin (83.5–90°N between 50–180°W and 78–90°N between 141–180°E), the Eurasian Basin (82–90°N between 30°W–140°E and 78–82°N between 110–140°E), and the Canada Basin (72–84°N and 130–155°W)–see Fig. 2." (line 168).

We agree to the Referee that this part of the text fits better in the introduction. We have moved this text to the introduction in lines 60-65.

-line 172: 'deeper than modelled ML'
Here, we are discussing the LOPS climatology and its tendency to exhibit excessively deep values along the coast of Greenland. This discrepancy may be attributed to the data obtained offshore, particularly in the ocean and deep ocean regions. We are not comparing it to the modeled ML.

-lines 176-178: It is hard to see these salinity differences from Figure 4 which shows absolute salinities. A solution is to add Control-LOPS salinity difference panels to Figure 4.
We thank the Referee for this suggestion. We have added the differences between the Control-LOPS for both salinity and temperature in Figure 4, as has been done for the MLD.

-line 181: 'data are averaged spatially and temporally'
We modified the text to (line 190): "Data is averaged spatially and temporally for each basin."

-line 189: Here is a place where indicating the parameter would be a reader: 'Increasing the fraction of surface TKE ($f_r$) that'
We thank the Referee for this suggestion. The revised text is (lines 199-200): "Increasing the fraction of surface TKE that penetrates into the ocean $f_r$ from 0 to 0.1 increases the MLD as well as the amplitude of the seasonal cycle."

-line 190: Do you mean 0.75 rather than 0.08?
We made a mistake in the higher value mentioned in the previous text. It should indeed be 0.1, not 0.08. The correction has been made, as shown in the previous response.

-lines 191-192: The effect of nn_tau is not shown in Figure 5, so this sentence in this context is a bit misleading.
The effect of the simulation $h_\tau = 10$ m (nn_tau) is shown in Figure 5 (in green). When the value of $h_\tau$ is changed from 30 m (in the control simulation) to 10 m in the $h_\tau = 10$ m simulation, the amplitude of the seasonal cycle decreases.

-line 194: 'could arguably be caused'
Included (line 203). The text have been modified to: "These summer biases could arguably be caused by the different reference depths used for computing the MLD with the density threshold criteria: 5 m for the LOPS climatology and 0.5 m for NEMO4.2."

-line 195: 'However, we recomputed'
It has been included as: "However, we recomputed the MLD using the 5m reference for the control run ..." (line 207).

-line 198: Is the MLD std modelled and/or regional, what are the summer months used? This information should be added.
We modified the text to: "The seasonal cycle of the MLD standard deviation during summer is almost negligible, and in winter, for the Makarov and Canada Basins, it remains below 15 m, showing a similar spatial variability between experiments in these regions (see Fig. A2 in Appendix)." (lines 211-213).

-line 201: 'and is only 8 m'
Done (lines 214-215). The new phrase is: "For instance, the MLD std reaches up to 30 m for the $\chi = 1$ experiment and is only 8 m for the $f_r = 0$ experiment."

Done (line 218). the new sentence is: "The largest ML deepening is observed for $\chi = 1$ (no attenuation of mixing due to sea ice coverage), with MDL at least 20 m thicker than the control run in both months across the studied regions. "

Done.

We included to the text as: "Conversely, the most significant ML shallowing is observed for $f_r = 0$ (TKE MLP turned off), resulting in ML shallower than the control by 20 m, particularly in the Canada and Eurasian Basins in March." (line 221).

We agree to the Referee and it has been corrected. The new sentence is: "Decreasing the characteristic depth of TKE penetration $h_\tau$ from 30 m to 10 m has an impact similar to a decrease of the penetrating fraction of energy $f_r$, with a similar spatial distribution in March and September." (lines 221-223).

We have changed this paragraph to the following: "A decrease in the MLD with a strong stratification corresponds to a reduction in sea surface salinity and an increase in surface temperature compared to the control simulation. In contrast, an increase in MLD with a weak stratification is associated with an increase in sea surface salinity and a decrease in surface temperature. This can be attributed to the fact that a shallower mixed layer and a strong stratification result in less mixing during ice melt, leading to a fresh anomaly at the surface and trapping heat at the surface. On the other hand, a deeper mixed layer and a weak stratification allow freshwater to mix more deeply, resulting in higher surface salinity and facilitating vertical heat exchange." (lines 263-269).

We thank the Referee for this suggestion. We have added the following clarification: "While our control simulation demonstrates improvements compared to these models, adjusting the TKE MLP parameters does not improve the representation of the temperature maximum below 200 m, as this maximum is primarily affected by heat advection at that depth." (lines 247-249).

The negative biases of the sea ice thickness are discussed in lines 281-282 as follows: "In March, negative biases between the simulated sea ice thickness and PIOMAS are more pronounced in the region next to Greenland. "

Indeed, we have clarified this part of the text by adding: "Specifically, regions near the East coast (Chukchi, East Siberian, Laptev, Kara, and Barents Seas) display sea ice thickness close to zero during this month." (line 280)

Included (line 281). The new sentence is: "In March, negative biases between the simulated sea ice thickness and PIOMAS are more pronounced in the region next to Greenland. "

The OSI-SAF uncertainty, as indicated by Lavergne et al. (2019), is approximately 3%. In the same study, a comparison with the ESA CCI data revealed uncertainties of similar magnitude. We opted not to include this dataset in our analysis because it only covers the period from 2002 to 2017.

We thank the Referee for this question. In winter, the central Arctic Ocean is almost completely covered

by sea ice, with concentrations reaching nearly 100%, and sea ice thickness ranging between 2 and 3 meters. During this period, the role of ice-albedo feedback in upper ocean-sea ice interactions is likely to be less significant due to the extensive ice cover. However, in summer, as sea ice melts and some regions are exposed to open water, the effect of ice-albedo feedback becomes more pronounced. As surface waters warm due to the increased absorption of solar radiation, ocean stratification increases. The warmer, less dense surface layer becomes more stable and less likely to mix with the cooler, denser water below. This enhanced stratification acts as a barrier, preventing the upward mixing of colder water which could otherwise help mitigate surface warming. Consequently, the ice-albedo feedback strengthens the link between stratification and sea-ice melt, creating a positive feedback loop that further accelerates ice loss.

-line 299: Would be reader friendly to remind what these three sensitivity experiments are, not just list their parameter values. Here Tables 1 and 2 could be used/ cited.
We have simplified the name of the experiment, and the table 2 have remove.

-Figure 11. Differences seem often not very clear and there can be biases both ways. Perhaps calculating RMS or mean absolute differences per basin would give clearer and more quantitative measure.
Figure 11 illustrates the spatial distribution of differences in sea ice concentration and thickness between the sensitivity experiments and the control run for March and September. The differences were calculated as the difference between the experiment and control for sea ice concentration. A positive difference indicates that the sea ice concentration in the control run is larger than in the experiment, while a negative value indicates the opposite. Additionally, the mean absolute differences are discussed in lines 300-310.

-Figure 12: Caption: 'MLD, sea-ice concentration and sea-ice thickness'. How much these timeseries correlate? Seems that year-to-ear variability is captured rather well due to the same atmospheric forcing and free drift in summer.
For a given experiment, there exists correlations between sea-ice concentration, MLD and sea-ice thickness. As explained in the last paragraph of section 3, correlations between MLD and sea-ice properties are explicitly mediated through the parametrization employed for the attenuation coefficient. In presence of a non-trivial attenuation coefficient, all time series tend to display similar trends. When the attenuation is turned off, though, ($\chi = 1$, purple timeseries) the MLD seems to decorrelate to the sea-ice variability.

-line 317: 'is close to'
Done (lines 331-332). We revised the phrase to "In the Canada Basin, the sea ice thickness is close to the PIOMAS reanalysis during the full period."

-line 325: What is short-term?
Thank you for the remark. Indeed, we referred to the short-term trend as the trend from 2000 to the present. We have changed the text to: "This short-term trend (from 2000 to the present) is also evident in the simulated sea ice thickness and the sea ice thickness from the PIOMAS reanalysis in all three basins." (lines 338-340).

-line 332: 'exhibit closer resemblances'
Included (lines 345-346). The new text is: "Comparing simulations with ITP observational data, the $\chi = 1 - f_i$ and control simulations exhibit closer resemblances."

-line 346: Can you link the MLD increase to the erosion of the Arctic halocline? We thank the Referee for this question. Indeed, as discussed in the previous section of the paper, there is a close link between the deepening of the ML and the Arctic halocline: a deeper ML is associated with weaker stratification, while a shallower ML exhibits a strong stratification. However, it is important to note that one is not the direct cause of the other. Including a detailed discussion on stratification in this section would have been repetitive. Therefore, we decided to omit it from this part of the paper.

-line 370: 'in these regions than the $1 - f\_i$ option'
Done (line 376).

-line 371: 'the LOPS climatology'
Done (line 377).

-line 372: 'upper ocean vertical properties'
This part of the text has been removed.

-line 398: 'their coverage'
Done (line 385).

-line 401: 'to entirely remove TKE MLP'
Done (line 388).

-line 405: 'this parameterization lacks a'
Done (line 410).

-lines 406-407: Do you have a reference for the claim of time dependence of the NEMO TKE MLP scheme? This reference to the time dependence was too technical for the paper, so we decided to remove it. However, the reference for this issue is as follows: `http://forge.ipsl.jussieu.fr/nemo/ticket/2748#no2`.

---

## Author Comment (AC2)

July 1, 2024

Sofía Allende
ELIC Place Louis Pasteur 3/L4.03.08
1348 Louvain-la-Neuve Belgique
Phone: +(33) 641842014
Email: sofia.allende@uclouvain.be

This studies tests a general modification to turbulence parameterization that redistributes turbulent kinetic energy across the base of the mixed layer to balance existing underestimations of the mixed layer depth, with respect to its performance on surface ocean and sea ice properties in the Arctic by evaluating multi-year runs with different parameter settings. Turbulence parameterizations are an important and critically underconstrained factor in ocean and climate models, and their improvement is vital to reduce uncertainties in climate predictions. The methods are established and sound, and the paper will be a good and important contribution to the improvement of ocean models.

Prior to publication, I believe the following major concerns would need to be addressed:

We thank the Referee for his/her careful report. We have followed all of his/her recommendation and we believe that this has significantly improved the quality of the manuscript.

Hereafter, we give a short list of the implemented changes in view of the Referee's comments. Please find below a point-by-point reply.

(1) Throughout the manuscript, the authors use both the assigned variables (chi, f_r, h_T) as well as the NEMO-internal identifiers (rn_efr, nn_htau etc) to refer to the paramters. This makes the manuscript very hard to follow, and gives the impression of a NEMO-specific technical report rather than a scientific paper. I strongly recommend removing all NEMO-internal identifiers and only use the actual variables. (If needed, a table could be added in the Appendix to relate to the NEMO identifiers, but I assume these can be found in a NEMO-specific documentation. I would also try to reduce the references to NEMO specific models to a minimum, and mention other model frameworks that use TKE MLP to make the study more appealing for readers outside the NEMO community.
We thank the Referee for pointing out the lack of clarity in our parameter references. We have revised the manuscript for consistency by removing all NEMO-internal identifiers and using only the actual variables: $\chi$, $f_r$ and $h_\tau$.

(2) In the results section, surface ocean properties are discussed individually, but these values are directly linked in a very straightforward and known way (more mixing = redistribution of salt = deeper ML = higher surface salinity = lower stratification). This separation leads to repetition and makes it hard to follow the paper. I recommend re-arranging both results sections to present findings on the individual diagnostics (MLD, surface salinity, stratification) side by side.
We have reorganized this section to enhance clarity and flow. We first introduce the mixed layer analysis, followed by the discrepancies in ocean stratification, salinity, and temperature vertical profiles, and end with the results on sea surface salinity and temperature.

(3) It would be great to have a table summarizing which configuration performs good/intermediate/bad for specific regions (the three basins) and processes (seasonal cycle, interannual trends, MLD; surface salinity, sea ice).
We have included a table in the discussion and conclusion section that summarizes the performance of

the MLD and sea ice thickness seasonal cycle and inter-annual variability for all regions studied: the Makarov, Eurasian, and Canada Basins.

(4) The overwhelming part of the discussion and conclusion section is a repetition of the results. The summary should be massively shortened, to one paragraph, and the discussion should address questions like: Why do the different configurations yield the observed difference in simulated conditions? What is your recommendation for the 'best' configuration of the TKE MLP in Arctic Environments? How much better does your identified best choice perform against the state-of-the-art configuration? Is it generally applicable, or do users need to discriminate which configuration to use based on regions? What are the limitations of the parameterization, ie, where does it not perform well? What data is needed to make further improvements? Is this parameterization the way to go ahead, or are there alternatives which might be superior in the long run?
We have revised this section to clarify our message. The table mentioned above has been included to highlight which simulations obtain better results in each region. We agree with the referee that some recommendations should improve the manuscript. Accordingly, we have incorporated these suggestions and discussed the limitations of our results.

Minor concerns:

- The description of the implemented underlying turbulence parameterization (not MLP) used in the methods section (l. 60-71) is very brief. For a manuscript focussing on improving turbulence parameterization, I would appreciate a more detailed description without having the reader go back to the original literature.
We have included the prognostic equation that describes the default TKE formulation in NEMO (lines 77-80):

The prognostic equation is given by:
$$\frac{\partial \bar{e}}{\partial t} = K_m \left( \frac{\partial \bar{U}_h}{\partial z} \right)^2 - K_\rho N^2 + \frac{\partial}{\partial z} \left( K_e \frac{\partial \bar{e}}{\partial z} \right) - \varepsilon$$

It results from the balance between the vertical shear, the dissipation of TKE due to buoyancy, the vertical diffusion of TKE, and the energy dissipation.

- l. 95 I do not understand what "excluding salinity restoration under sea ice" means, please add a brief explanation.
In the NEMO model, there is an option for surface restoring to observed Sea Surface Temperature (SST) and/or Sea Surface Salinity (SSS). The SSS restoring term serves as a flux correction on freshwater fluxes to mitigate uncertainties in the observed freshwater budget. In our simulation, this option is deactivated. We have included the following explanation in line 109: "In our simulations, we configure the setup to exclude salinity restoring under sea ice. This means that no flux correction on freshwater fluxes was applied."

- l. 96 Rathore et al. - year is missing
We added the year to the reference.

- Table 1 is not necessary, is / can be incorporated in the text
We have modified the table to include the values of each parameter. We believe it is useful for the reader to have this information summarized.

- Ocean and Sea Ice variables around l. 115: No need to give the NEMO-internal identifiers, stick to the commonly used symbols.
We have removed the NEMO-internal identifiers in the new text (lines 128-132).

- l. 125. The density threshold criterion is repeated in two different sentences
We have rewritten this part, and the information is now presented in a clear and concise manner. Here is the revised text with the correction (lines 139-141): "The MLD is calculated for each individual profile by employing the threshold density criterion. This criterion is based on the difference in density between a given depth ($z$) and a reference depth ($= z_{ref}$), which is denoted as $\Delta \rho = \rho(z) - \rho(z_{ref})$. The MLD

is then defined as the depth where this difference in density exceeds the threshold value of 0.03 $kg/m^3$ (e.g. de Boyer et al. (2004)).”

- l. 127 “The choice of the reference depth is impactful” - please provide details on why it is, and also I do not understand why you pick different reference depths for model / obs later on, what was the reason for that (especially as you state the choice does NOT make a difference, around l. 195)
We have added an explanation to clarify this issue. In the study by Treguier et al. (2023), it was shown that changing the reference depth of the model CMCC-NEMO from 0.5 m to 10 m can lead to differences in the MLD exceeding 40 m across large areas of the Southern Hemisphere oceans. In the Arctic region, these differences are smaller, reaching only a few meters, as we have also demonstrated using our results and comparing a reference depth of 0.5 m to 5 m. Specifically, during the summer months of July and August, these differences are less than 3 meters. While these variations are significant given the shallow MLD values during these months, they alone cannot account for the discrepancies observed with the LOPS climatology for the same periods.

The new version is (lines 204-207): “As shown by Treguier et al. (2023), the reference depth significantly impacts the MLD. For instance, altering the reference depth in the CMCC-NEMO model from 0.5 m to 10 m leads to differences in MLD exceeding 40 meters across large areas of the Southern Hemisphere oceans. In the Arctic region, these differences reach a few meters.”

- if nighttime convection is not an important issue in the Arctic, shorten the corresponding paragraph
We have shortened the paragraph to (lines 142-147): “The LOPS climatology estimates the MLD as the depth mixed over at least one daily cycle, typically assumed to be no less than 10 m deep (e.g., Brainerd et al. (1995)). This depth filters out possible daily stratification in the top few meters, which is common in the tropics or summer mid-latitudes. However, in the Arctic Ocean, where the diurnal cycle linked to solar heat fluxes is minimal or nonexistent, especially with ice present, the MLD can be shallower than the usual 10 m depth (e.g., Peralta et al .(2015)). Therefore, we have recomputed the MLD for the Arctic region using a more appropriate reference depth of 5 meters.”

- Table 2. Needs improvement, I do not understand it.
This table has been removed due to changes in the reference of the simulations. We no longer use the internal NEMO names for the variables.

- l. 137. A citation is needed to the TEOS-10 part, no need to mention a python toolbox was used. Also, consistently using TEOS-10 would make it consequent to use absolute salinity and not practical salinity (which is unitless, not pss) throughout the manuscript. The TEOS-10 framework also includes the Brunt-Väisäila frequency N, which should be introduced here instead of around l. 240 in the results, with a shorter description.
We added a citation for TEOS-10. We agree that our previous text was confusing and not clear about why we talked about the conservative temperature and absolute salinity profiles. Initially, we intended to indicate that vertical potential density profiles were computed using the toolbox that uses such quantities. However, this is unnecessary and adds confusion, especially since our results are based on practical salinity and potential temperature. Therefore, we removed this part, and the revised text is (lines 149-151): “We computed vertical potential density profiles using the TEOS-10 Gibbs Sea Water toolbox (Mcdougall et al (2011), and determined the MLD by applying the threshold density criteria. This was done to compare ITP observational data with NEMO sensitivity experiments, where the surface reference depth for ITP varies from 10 to 0 m depending on the profile.”

- l. 157-162: you introduce upper ocean and sea ice variability by describing the seasonal cycle, but there are also huge regional differences in the Arctic Ocean that should be summarized here.
The description of the seasonal cycle has been moved to the introduction. We have incorporated regional differences as follows (lines 54-60): “The Makarov Basin, located north of Siberia, experiences seasonal ice cover and receives freshwater from the East Siberian Sea. Shallow depths (500-1500 m) render it highly sensitive to sea ice variability and freshwater inputs. The Eurasian Basin extends from the Siberian Shelf to the North Pole, featuring extensive multi-year ice cover and significant freshwater discharge from major Arctic rivers. Depths can reach 4000 m, influencing Arctic freshwater storage and sea ice dynamics. The Canada Basin, situated between North America and Siberia, is dominated by multi-year ice influenced by the Beaufort Gyre. Its complex bathymetry, including deep ridges like Alpha and Mendeleev, affects ocean circulation patterns and carbon cycling.”

- l. 169: "MLD discrepancies are less pronounced" compared to other NEMO simulations, or any other simulations in general?

We have clarified the text as follows (lines 175-177): "Compared to a large portion of global models forced by CORE-II and JRA55-do, as studied by Ilicak et al. (2016) and Allende et al. (2023), which include both NEMO and non-NEMO models, MLD discrepancies with observational data are less pronounced."

- l. 173: "This would be one point to be improved in a future version of this data set" belongs into the discussion.

We have significantly revised the discussion according to the recommendations of the referee. We have however kept this specific sentence in this part of the text to emphasize the difference between the control run and the LOPS climatology.

- l. 195 (and around): Confusing, why not just use the same reference density? Also, move to methods.

This information is already detailed in the methods section. We utilize the MLD output directly from NEMO, which is consistent with the CMIP6 protocol. This approach enables straightforward comparisons with other models and ensures higher accuracy compared to monthly density profile recalculations. Here, we examine the implications of this approach, particularly regarding the shallow MLD values observed in the summer months. We recomputed the MLD using the same reference depth as the LOPS climatology and found discrepancies of up to 3 meters between the two methods. However, these differences are insufficient to fully account for the discrepancies observed with the LOPS climatology.

- I am not sure if I understand the meaning of the standard deviation comparison: Is that a measure for the internal variability in the three basins? Please expand.

Your understanding is correct. The standard deviation associated to the depth of the mixed layer quantifies its spatial variability in each basin. We have added the sentence "To quantify the spatial variability of the mixed layer within each basin, we measure the MLD standard deviation for each month"

- l. 214-216: Not only sea ice melting affects salinity anomaly, also advection of (modified) river water!

We have modified and moved the text to improve clarity. The new text reads as follows (lines 262-269): "The discrepancies between the sensitivity experiments and the control run for the spatial distribution of sea surface salinity and sea surface temperature exhibit a similar pattern (see Fig. 8). A decrease in the MLD with a strong stratification corresponds to a reduction in sea surface salinity and an increase in surface temperature compared to the control simulation. In contrast, an increase in MLD with a weak stratification is associated with an increase in sea surface salinity and a decrease in surface temperature. This can be attributed to the fact that a shallower mixed layer and a strong stratification result in less mixing during ice melt, leading to a fresh anomaly at the surface and trapping heat at the surface. On the other hand, a deeper mixed layer and a weak stratification allow freshwater to mix more deeply, resulting in higher surface salinity and facilitating vertical heat exchange."

- l. 219-230: This comparison to other model runs feels out of place here, but could be modified and go into the discussion. Also, Atlantic Water temperature at 200m depth should be relatively unaffected by modifications of the surface(!) mixing parameterization, right?

Related to the Atlantic Water temperature at 200m depth, we have added the following clarification (lines 247-249): "While our control simulation demonstrates improvements compared to these models, adjusting the TKE MLP parameters does not improve the representation of the temperature maximum below 200 m, as this maximum is primarily affected by heat advection at that depth."

- Figures 7, 8: specify which temperature is shown (potential, conservative)?

We have added to Figures 7 and 8 that we are using potential temperature, as mentioned in the methods section. This should help clarify any confusion.

- l. 239ff. Higher turbulence reduces stratification is quite basic - I am not sure if this paragraph adds to the story of the paper. Consider to remove?

We apologize for any confusion, but line 239 does not contain a reference to "Higher turbulence reduces stratification." Therefore, we are unable to evaluate the suggestion to remove that paragraph. If there is a specific paragraph or section that the reviewer is referring to, please provide additional context or a specific line number for clarification.

l. 251- l. 257. I cannot follow the FWC part: Mixing only redistributes salt / FW, so is the difference from additional ice melt? The numbers seem high for that. Also, for FWC results, a discussion of the sensitivity to choice of reference salinity is needed.

We agree that the calculation of FWC is somewhat out of context in the paper. Therefore, we have decided to remove it. However, here is the explanation of the calculation:

We calculate the upper-ocean seasonal freshwater content (sFWC) by:

$$sFWC(t) = \int_{Z_{fw}(t)}^{0} \frac{S_{may} - S(t,z)}{S_{may}} dz,$$

where $S$ is the salinity at a month t, and $Z_{fw}(t)$ is the vertical extent of mixing approximated here by the MLD of May. Therefore, $S(t,z)$ is the salinity of the month z at its ML deepening. We use the May-average conditions as a reference salinity value following Rosenblum et al. (2021), different from the commonly used 34.8 ppt, to avoid errors stemming from arbitrary values and ensure a fair comparison between experiments Schauer et al. (2019).

l. 260: Add HOW sea surface properties affect sea ice.

We have included how sea surface properties can modify sea ice as follows (lines 272-274): 'For instance, higher sea surface salinity lowers the freezing point of seawater, delaying the formation of sea ice. Conversely, lower sea surface salinity raises the freezing point, promoting sea ice formation. Additionally, colder sea surface temperatures encourage sea ice formation, while warmer sea surface temperatures contribute to sea ice melt."

l. 266 what is meant by "East coast"

We have clarified this part of the text by adding (lines 280-281): "Specifically, regions near the East coast (Chukchi, East Siberian, Laptev, Kara, and Barents Seas) display sea ice thickness close to zero during this month."

l. 292 (and again later): remove "leads to a large Richardson number, which" - the Richardson number has never been defined here, relation stratification and Ri is trivial, so it adds no information here.

We agree with the Referee. The two references to the Richardson number have been removed.

l. 308. "unrealistic seasonal cycle" - unrealistic in which way? remove "displaying a nearly flat linear regression", that is the same as 'no trend'

We have incorporated the suggested changes to the text as requested. The revised text now reads (lines 321-322): "Consistent with Fig. 5, all experiments simulate a ML shallower than observations, except for the $\chi = 1$ experiment with strong TKE MLP under sea ice, which exhibits a too depth seasonal cycle."

l. 363. "beneficial in the NEMO model" - in which way and where? Open water?

We clarified this part as follows (lines 367-370): "As noted by Calvert et al. (2013), Rodgers et al. (2014) and Storkey et al. (2018), the additional source of mixing by TKE MLP is beneficial in the NEMO model to reach realistic MLD in the Southern Ocean and in open water regions, which we demonstrate holds true for the Arctic region as well (see Table. 2)."

l. 386 "biases are not due to vertical mixing" - what could be likely other reasons for these biases then?

We clarified this part as follows (lines 405-408): "The underestimation of sea ice thickness and sea ice concentration in the control run remains in all sensitivity experiments, which show that these biases are not due to vertical mixing only. One potential reason for this could be the ERA5 forcing, which introduce warmer temperatures in the Arctic. As shown by Batrak et al. (2019), ERA5 has a warm bias in winter, leading to thinner ice and a reduced summer extent in the model. Further investigation is needed to explore this aspect.

---

## Author Comment (AC3)

July 12, 2024

Sofía Allende
ELIC Place Louis Pasteur 3/L4.03.08
1348 Louvain-la-Neuve Belgique
Phone: +(33) 641842014
Email: sofia.allende@uclouvain.be

The present manuscript focuses on improving the choices that go into a mixing scheme used in the NEMO ocean model. The authors have evaluated a vertical mixing parameterization based on turbulent kinetic energy (TKE) in the Arctic region. They have checked the mixed layer penetration effects modeled in the TKE based scheme. The MLD penetration effects account for additional mixing caused by near inertial effects and ocean swells. They find that it impacts the evolution of mixed layers below sea-ice and impacts the formation of sea-ice itself. They also checked the influence of adding information about the sea-ice concentration in the mixing scheme and it leads to trends in the mixed layer depths that agree with observations.

The authors convincingly argue about the choices of tunable parameters used in mixing parameterizations. Unconstrained turbulence mixing parameterizations lead to major sources of uncertainty in climate projections. It is vital to improve mixing parameterizations, especially those of upper ocean processes which regulate the upper ocean response to surface forcing. Because of the important nature of the work done in this manuscript, I recommend this article to be published in the journal 'Geoscientific Model Development'.

We thank the referee for their detailed assessment of our work and positive appreciation. The points they raised were very useful in improving our work. Please find below detailed answers to each of their specific remarks.

I have a few minor comments / suggestions to improve the clarity of the manuscript:

1. Line 75: "The parameterization is activated in NEMO when the parameter nn_etau is set to 1 (and deactivated when nn_etau=0)."

This belongs in the NEMO documentation rather than in this article. If the authors want to document this, they can consider adding these kind of sentences to a supplementary material for additional documentation.
We have removed this phrase as it was too technical for the new version of the text.

2. Line 75: It would be helpful to write the evolution equation of TKE. It would help in understanding how the various terms such as $N^2$, velocity, and energy (e at time t) go into the evolution of e at time t+dt.
We have included the prognostic equation that describes the default TKE formulation in NEMO (lines 77-80) as follows:

" The prognostic equation is given by:

$$\frac{\partial \bar{e}}{\partial t} = K_m \left( \frac{\partial \bar{U}_h}{\partial z} \right)^2 - K_\rho N^2 + \frac{\partial}{\partial z} \left( K_e \frac{\partial \bar{e}}{\partial z} \right) - \varepsilon$$

It results from the balance between the vertical shear, the dissipation of TKE due to buoyancy, the vertical

diffusion of TKE, and the energy dissipation."

3. The equation written in line 75 is a bit confusing. On the left hand side, the authors have written e(t+dt,z), and on the right hand side is there a missing term: e(t,z) ?
Should the equation be:
$e(t + dt, z) = ((de(t, z)/dt) \times \delta t) + e(t, z) + e_{inertial}(t, z),$

Where all the d are partial derivatives?

Please check the equation in line 75.
We thank the referee for pointing this out. The expression taken from Calvert et al. (2013) is:

$$\bar{e}(t + \Delta t, z) = \int_t^{t+\Delta t} \left( \frac{\partial}{\partial t} \bar{e}(t, z) \right) + \bar{e}_{inertial}(t, z)$$

which we rewrote as:

$$\bar{e}(t + \Delta t, z) = \frac{\partial}{\partial t} \bar{e}(t, z) \Delta t + \bar{e}_{inertial}(t, z)$$

Indeed, the expression is a bit confusing, so we have decided to remove it. We believe that the explanation is better without it, as it was intended to be an ad hoc parameterization of kinetic energy production through a source term.

The phrase has been modified as follows: "The TKE $\bar{e}(t, z)$ includes an additional energy source term $\bar{e}_{inertial}(t, z)$, which represents the contribution of the TKE MLP as:"

4. Table 1: This table is not necessary. You have already given a short summary around line 100.
We have revised Table 1 to include the values of each parameter. We believe summarizing this information in a table format provides a useful and easy comparison across models for the reader.

5. Line 125: You have already defined MLD (0.03) at line 115.
We thank the referee for pointing this out. However, we believe it is necessary to retain the expression to introduce the values of the reference depth, which differ between NEMO (0.5m) and the LOPS climatology (5m).

6. Line 195: Do you mean standard deviation of MLD?
In line 195, we do not reference the standard deviation. We discuss the differences in the reference depth between NEMO and the LOPS climatology.

The discussion about the standard deviation is now around line 209 in the new version of the manuscript. The phrase has been modified as follows:

" To quantify the spatial variability of the mixed layer within each basin, we measure the MLD standard deviation for each month. The seasonal cycle of the MLD standard deviation during summer is almost negligible, and in winter, for the Makarov and Canada Basins, it remains below 15 m, showing a similar spatial variability between experiments in these regions (see Fig. A2 in Appendix). However, for the Eurasian Basin, differences between experiments appear to be more substantial. For instance, the MLD std reaches up to 30 m for the $\chi = 1$ experiment and is only 8 m for the $f_r = 0$ experiment."

7. Line 80: Using the symbol f_r should be enough for the manuscript. Describing the namelist parameter is not needed.
We thank the Referee for the suggestion. We have modified the full text to refer to the simulations using the names of the parameters in the equation ($f_r$, $\chi$ and $h_\tau$), rather than the individual names of the NEMO parameters (rn_efr, nn_eice and nn_htau).

8. Line 95: Please provide a reference for ORCA1. Is this a specific configuration or some different model is not clear.
We have included the following explanation: "To carry out this investigation, we utilize the NEMO4.2 version with the SI3 sea ice model, using the eORCA1 configuration. The eORCA1 quasi-isotropic global tripolar grid has a nominal resolution of $1°$, extended to the south to better represent the contribution of Antarctic under-ice shelf seas to the Southern Ocean freshwater cycle. The grid has a latitudinal grid

refinement of $1/3°$ in the equatorial region. The vertical discretization consists of 75 levels, where the initial layer thicknesses increase non-uniformly from 1 m at the surface to 10 m at 100 m depth, reaching 200 m at the bottom.'' (lines 103-107).

9. Line 130: For table 2, you could provide an experiment number as an additional column before the 'parameter' column. In general, I find that writing down both: variables and namelist variables used in NEMO code to be confusing. The authors could consider using only symbols rather than namelist variables. All the symbols and their corresponding namelist variables could be added to a table in an appendix.

Following the answer to remark 7, we have removed this table because we now refer to the simulations using the names of the parameters in the equation, rather than the individual names of the NEMO parameters.

10. Line 140: "The WOA23 dataset is available at the NOAA website." No need for this sentence. You can reference WOA in the earlier sentence.

We have removed this phrase. The revised text now reads: '' Our study also incorporates temperature and salinity vertical profiles provided by the latest version of the World Ocean Atlas 2023 (WOA23), which integrates data from 1955 to 2022 at a resolution of $1°$ (Regan et al. 2023). ''

11. Around lines 195-205: These sentences seem more suitable in the discussion.

We have extensively revised the discussion section. However, we chose to keep this particular sentence in this part of the text to highlight the differences between the control run and the LOPS climatology.

12. Line 240: When you mention the Brunt-Vaisala frequency, please state to refer to results in Figure 8.

We have updated this part of the manuscript. The new text now begins the discussion of upper ocean properties with the Brunt-Vaisala frequency, as follows (lines 226-228): "Fig. 7 shows the vertical distribution of ocean physical properties in September, including stratification, salinity, and temperature. To assess the stratification strength across simulations, we calculate the Brunt-Väisälä frequency as ..."

13. Line 370: 'Surprising' seems like a strong word. Stratification is sensitive to OSBL mixing parameterization, so it is not that surprising. Just remove the word "surprising".

We have thoroughly revised the conclusion and discussion section, and this phrase has been removed.

---

## Author Response (AR2)

August 28, 2024

Sofía Allende
ELIC Place Louis Pasteur 3/L4.03.08
1348 Louvain-la-Neuve Belgique
Phone: +(33) 641842014
Email: sofia.allende@uclouvain.be

Dear Editor,

Thank you for your positive feedback on our revised manuscript. We are pleased that the reviewers are satisfied with the corrections and responses we have made.

We have now implemented the minor corrections provided by the reviewers, as detailed below.

We appreciate the opportunity to publish our work in GMD and we thank the referees for their constructive feedback.

Best regards,

Sofia Allende
* * *
**Referee 1:**

I thank the authors for addressing my questions and critical points well. I think the manuscript is now almost ready for the publication. I noted only three very small things:

- Would be good to mention that the upper limit of $f_r$ is 0.1 for example in line 94.
We changed the phrase to: " Here, z is the depth, $f_r$ is the fraction of the surface TKE $e_{surf}$ that penetrates into the ocean, and it ranges from 0 to a maximum of 0.1."

- line 94: remove comma after $f_r$
Ok.

- line 188: 'Data are'
Ok.

- There is no discussion on the mean absolute error in lines 293-303, or elsewhere in the manuscript.
As previously discussed, Figure 11 illustrates the spatial distribution of differences in sea ice concentration and thickness between the sensitivity experiments and the control run for March and September. The differences were calculated as the difference between the experiment and control for sea ice concentration and sea ice thickness. For example, a positive difference indicates that the sea ice concentration in the control run is larger than in the experiment, while a negative value indicates the opposite. For instance, in the referenced lines, we write:
" The simulation with no TKE MLP ($f_r = 0$) displays a 7% increase in sea ice concentration relative to the control case, while simulations with more TKE MLP under sea ice ($\chi = 1$ and $\chi = 1 - f_i$) show a 10% and 7% decrease in September, respectively. A similar behavior is observed for sea ice thickness, with

the largest differences between experiments observed in the Canada Basin. In September, in the $f_r = 0$ experiment, sea ice thickness increases by 14 cm; in the $\chi = 1$ experiment, sea ice thickness decreases by 24 cm; and the $\chi = 1 = f_i$ experiment shows a decrease of 13 cm in the sea ice thickness, relative to the control case."

**Referee 2:**

This paper examines the impact of ocean vertical mixing on sea-ice and upper ocean properties in 1 degree NEMO-SI3 simulations. It looks like this paper has gone through several rounds of review previously and thus is in very good shape, even if this is my first look at it. At this stage, I only have minor technical comments as the latest version reads well, with good quality figures.

- L80: Define what you mean by the last version of NEMO. Given many groups use different versions, explicitly mention the version in question.
We thank the Referee for this remark. We have revised the phrase to avoid referencing the NEMO version, as it is explained later. The updated text now reads: " Madec et al. (2017) introduce significant modifications to this parameterization."

- L106: Given NEMO is a geopotential coordinate model, I think level thickness is a more appropriate term than layer thickness.
Ok.

- L108: What do the authors mean by "atmospheric forcing for the ocean"? Is it different for the sea-ice? Or do the authors just mean all the NEMO experiments in this paper are forced by ERA5?
Yes, this phrase simply means that all the NEMO experiments are forced by ERA5. We have changed the text to: " In our simulations, NEMO is forced by ERA5 reanalysis (hersbach et al. (2020))."

- Figure 2: The yellow dashed line for the Canada Basin box is near impossible to sea. Even the green for the Eurasian Basin is not very clear. Use other colours.
We have modified Figure 2 to provide a better visualisation of the three basins.

L216: MLD (not MDL).
Ok.

- Figure 8 caption: What does "maps difference" mean? Maybe an of is missing after maps?
Ok.

- L320: deep not depth.
Ok.

- L405: In the discussion, the authors mention other ice covered regions. I'd like to see some speculation about which parameterization to use for these areas. Additionally, as the climate warms, and sea-ice decreases, might that require changes to the parameterization in the future when there is less ice?
We thank the referee for this suggestion. We have added the following phrase: " In particular, the Arctic region has been significantly impacted by global climate change, resulting in a rapid decrease in sea ice extent. This phenomenon is expected to alter the exchanges between the atmosphere and ocean, thereby affecting the fully-coupled ice-air-ocean system in the Arctic, and consequently influencing the mechanisms driving the TKE MLP parameterization." (lines 416-419).